# The distinct roles of calcium in rapid control of neuronal glycolysis and the tricarboxylic acid cycle

Carlos Manlio Díaz-García*, Dylan J Meyer, Nidhi Nathwani, Mahia Rahman, Juan Ramón Martínez-François, Gary Yellen*

Department of Neurobiology, Harvard Medical School, Boston, United States

**Abstract** When neurons engage in intense periods of activity, the consequent increase in energy demand can be met by the coordinated activation of glycolysis, the tricarboxylic acid (TCA) cycle, and oxidative phosphorylation. However, the trigger for glycolytic activation is unknown and the role for $Ca^{2+}$ in the mitochondrial responses has been debated. Using genetically encoded fluorescent biosensors and NAD(P)H autofluorescence imaging in acute hippocampal slices, here we find that $Ca^{2+}$ uptake into the mitochondria is responsible for the buildup of mitochondrial NADH, probably through $Ca^{2+}$ activation of dehydrogenases in the TCA cycle. In the cytosol, we do not observe a role for the $Ca^{2+}$/calmodulin signaling pathway, or AMPK, in mediating the rise in glycolytic NADH in response to acute stimulation. Aerobic glycolysis in neurons is triggered mainly by the energy demand resulting from either $Na^+$ or $Ca^{2+}$ extrusion, and in mouse dentate granule cells, $Ca^{2+}$ creates the majority of this demand.

## Introduction

Energy demand in neurons is proportional to the frequency of action potentials, which can increase dramatically upon neuronal stimulation. During neuronal activity, $Na^+$ and $Ca^{2+}$ enter the cell via specific voltage-gated ion channels in the plasma membrane. This, in turn, accelerates $Na^+$ and $Ca^{2+}$ pumping and leads to higher ATP consumption, which is replenished by oxidative phosphorylation (OXPHOS) and glycolysis (*Attwell and Laughlin, 2001*; *Yu et al., 2018*). However, whether the upregulation of ATP synthesis is provoked simply by the degradation of ATP (and increases in ADP), or by a feedforward signal such as elevation of intracellular $Ca^{2+}$ or AMP, is still not fully understood. (AMP can act as a feedforward signal via AMPK activation [(*Herzig and Shaw, 2018*)], or, like ADP, it can directly activate the glycolytic enzyme phosphofructokinase [(*Passonneau and Lowry, 1962*)].)

Both as an energy burden and as a feedforward signal, intracellular $Ca^{2+}$ is a strong candidate for coordinating the fast-metabolic responses to increased neuronal activity. Calcium handling is metabolically expensive and compartmentalized, involving several transporters at the plasma membrane and the endoplasmic reticulum. In addition, $Ca^{2+}$ uptake into the mitochondria can directly dissipate the mitochondrial membrane potential (*Duchen, 1992*; *Berndt et al., 2015*) and increase OXPHOS while activating several dehydrogenases in the tricarboxylic acid (TCA) cycle (*Denton et al., 1972*; *Denton et al., 1978*; *McCormack and Denton, 1979*; *Denton, 2009*; *Wescott et al., 2019*). When elevated in the cytosol, $Ca^{2+}$ can also modulate the activity of enzymes through $Ca^{2+}$/calmodulin-dependent signaling (*Singh et al., 2004*; *Marinho-Carvalho et al., 2009*; *Schmitz et al., 2013*), or direct allosteric regulation of cytoskeletal interactions (*Chen-Zion et al., 1993*).

Using genetically encoded fluorescent biosensors, NAD(P)H autofluorescence imaging, and extracellular $O_2$ measurements, we investigated the effects of mitochondrial $Ca^{2+}$ uptake on NADH production and consumption via the TCA cycle and the electron transport chain, respectively. Since

*For correspondence:
carlos_diazgarcia@hms.harvard.edu (CMD);
gary_yellen@hms.harvard.edu (GY)

NADH production in the cytosol is also increased upon acute neuronal stimulation, we tested if several $Ca^{2+}$ signaling pathways, or $Ca^{2+}$ itself, were necessary for this response. Finally, we assessed whether the energy demand resulting from $Na^+$ or $Ca^{2+}$ extrusion could trigger aerobic glycolysis in stimulated neurons.

## Results

### Fast cytosolic NADH responses are independent of mitochondrial NADH responses

Eukaryotic cells use several core metabolic pathways for energy metabolism that are compartmentalized between the cytosol and mitochondria (diagrammed in *Figure 1*). To monitor the metabolic changes that occur in response to neuronal stimulation, we measured the NADH/NAD$^+$ ratio in the cytosol using Peredox (*Hung et al., 2011*), which reflects the overall activity of glycolysis. We also imaged mitochondrial NADH in a population of dentate granule cells (DGCs) of the mouse hippocampus using UV-excited autofluorescence (*Chance et al., 1962*). This is known as the NAD(P)H signal because of a small contribution of the spectrally indistinguishable cofactor NADPH. In neurons, this signal is thought to predominantly reflect NADH changes in the mitochondria, because both the NADH concentration and the fraction of brighter protein-bound NADH, are higher in the mitochondrial matrix compared to the cytosol (reviewed by *Kann and Kovács, 2007*; *Shuttleworth, 2010*; *Yellen, 2018*). We also expressed the $Ca^{2+}$ sensor RCaMP1h (*Akerboom et al., 2013*) in the cytoplasm of DGCs to simultaneously monitor neuronal activity.

The NADH pool in one compartment (e.g. mitochondria) can potentially influence the NADH pool in the other (e.g. cytosol) due to their connection via the malate-aspartate shuttle (MAS), whose components are highly expressed in neurons (reviewed by *McKenna et al., 2006*). Therefore, we first tested whether the NADH signals in the cytosol could occur independently from the responses in the mitochondria.

The production of NADH in the mitochondria is preferentially fueled by pyruvate, which can be produced either from glycolysis or from lactate re-oxidation (via the lactate dehydrogenase reaction). Pyruvate is transported into the mitochondria via the mitochondrial pyruvate carrier (MPC) (*Bricker et al., 2012*; *Herzig et al., 2012*), where it is converted to acetyl-CoA by the enzyme pyruvate dehydrogenase and then further catabolized in the TCA cycle. We hypothesized that by inhibiting the MPC, pyruvate should fail to enter mitochondria and fuel the TCA cycle, and, as a result, the mitochondrial NAD(P)H signals (but not glycolysis) should be attenuated and pyruvate should accumulate in the cytosol.

We first confirmed that MPC inhibition leads to cytosolic pyruvate accumulation by expressing the pyruvate-sensitive FRET sensor Pyronic (*San Martín et al., 2014*) in the cytosol of DGCs. When slices were treated with a high-affinity inhibitor of the MPC (UK5099; 2 µM), the sensor reported a steady accumulation of cytosolic pyruvate over 30 min, with a signal comparable to that produced by exogenous addition of 10 mM pyruvate to the bathing solution (*Figure 2*, *Figure 2—figure supplement 1*).

Starving the mitochondria of pyruvate should also hamper the activity of the TCA cycle, thus diminishing NADH production in the mitochondria. To test this hypothesis, we assessed the effect of UK5099 on the NADH dynamics in the mitochondria of stimulated neurons.

In control conditions, we observed the typical NAD(P)H signals in response to electrical stimulation of DGC axons, consisting of a prompt and brief (1–2 s) negative deflection ('dip'), followed by an 'overshoot' that can last several minutes (*Figure 2b* Left; *Shuttleworth et al., 2003*; *Brennan et al., 2006*; *Brennan et al., 2007*; *Ivanov et al., 2014*). The dip reflects an oxidative phase, triggered by the dissipation of the electrochemical proton gradient across the inner mitochondrial membrane ($\Delta\mu_{H+}$) by $Ca^{2+}$ and/or ADP uptake into the matrix (*Duchen, 1992*; *Berndt et al., 2015*; reviewed by *Yellen, 2018*). The subsequent overshoot (reductive phase) is the result of NADH production by several dehydrogenases in the TCA cycle, but while some studies suggest that it is activated by mitochondrial $Ca^{2+}$ (*Duchen, 1992*; *Kann et al., 2003*), others dispute this for a variety of reasons, often involving measurements in different conditions (*Shuttleworth et al., 2003*; *Kasischke et al., 2004*; *Baeza-Lehnert et al., 2019*).

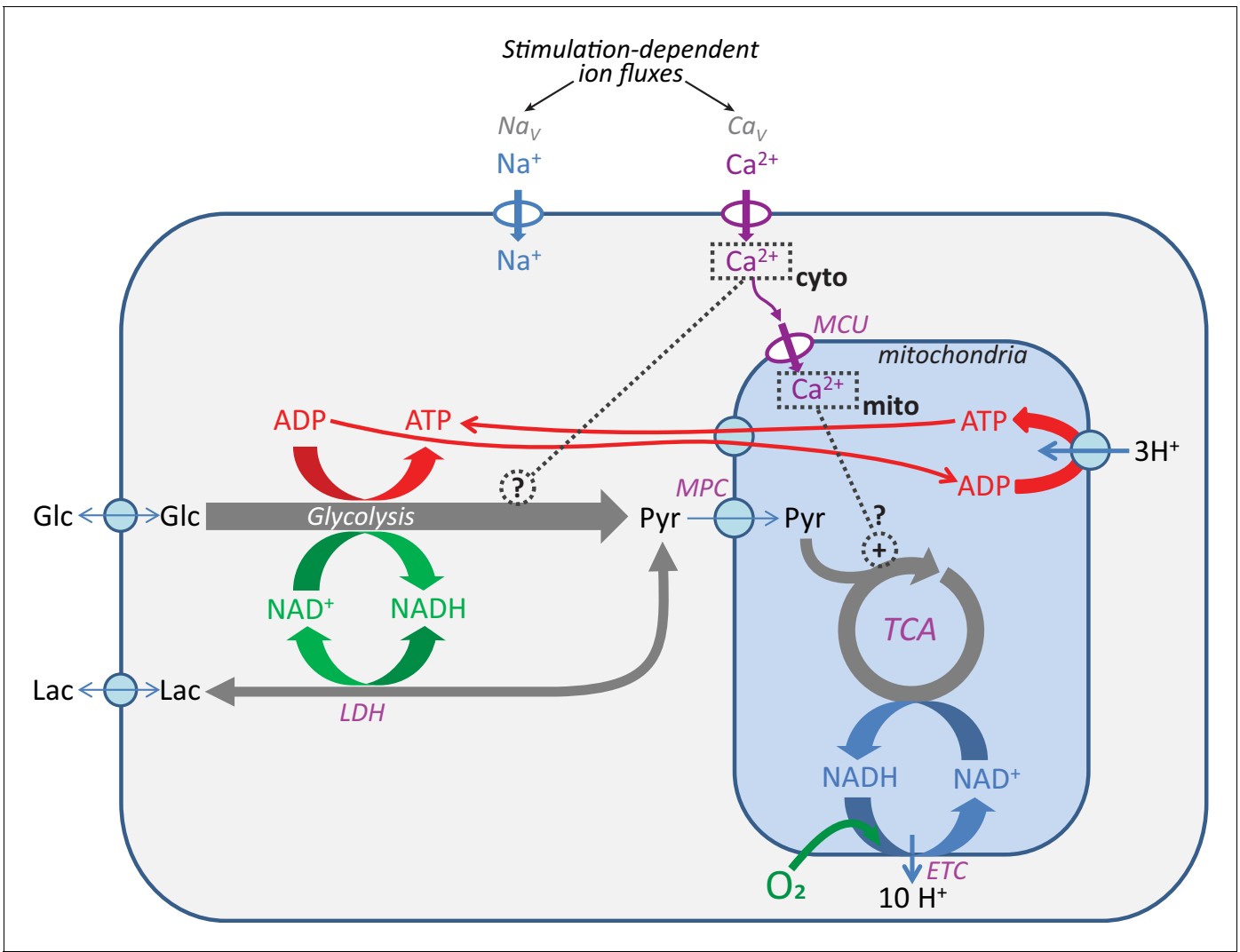

**Figure 1.** Pathways for core energy metabolism and the possible influences of calcium. Glucose (Glc) and lactate (Lac) are possible fuel molecules, and both the ATP/ADP and NADH/NAD$^+$ pairs are compartmentalized between cytosol and mitochondria, as is [Ca$^{2+}$]. For mitochondria, the mitochondrial calcium uniporter (MCU) and mitochondrial pyruvate carrier (MPC) are shown, as well as the tricarboxylic acid (Krebs) cycle (TCA) coupled to oxidative phosphorylation (OXPHOS), performed by the electron transport chain (ETC), the proton gradient, and the ATP synthase. Voltage-gated Na$^+$ channels (Na$_V$) and voltage-gated Ca$^{2+}$ channels (Ca$_V$) are the main pathways for ion entry to the neuronal somata during neuronal excitation. LDH = lactate dehydrogenase.

The online version of this article includes the following figure supplement(s) for figure 1:

**Figure supplement 1.** Ca$^{2+}$ regulation of the mitochondrial NADH shuttles.

**Figure supplement 2.** How ion influx influences cytosolic energy state.

MPC inhibition severely impaired NADH production in the mitochondria, both at baseline (*Figure 2—figure supplement 2*) and especially during periods of intense neuronal activity. Stimulation of the DGCs revealed a dramatic ~7-fold reduction of the overshoot in the NAD(P)H signal (*Figure 2b* Right; *Figure 2—figure supplement 3b*). MPC inhibition also steadily decreased the baseline NAD(P)H signal by 8.3 ± 2.9% from the initial values after ~35 min (*Figure 2b* Left, *Figure 2—figure supplement 2b*), reflecting an overall decrease in the baseline activity of the TCA cycle. Consistent with this, baseline O$_2$ levels increased by 25 ± 16%, indicating that O$_2$ consumption at baseline was also diminished in the presence of UK5099 (*Figure 2—figure supplement 2c*), probably because mitochondrial NADH fuels most of the O$_2$ consumption by the electron transport chain.

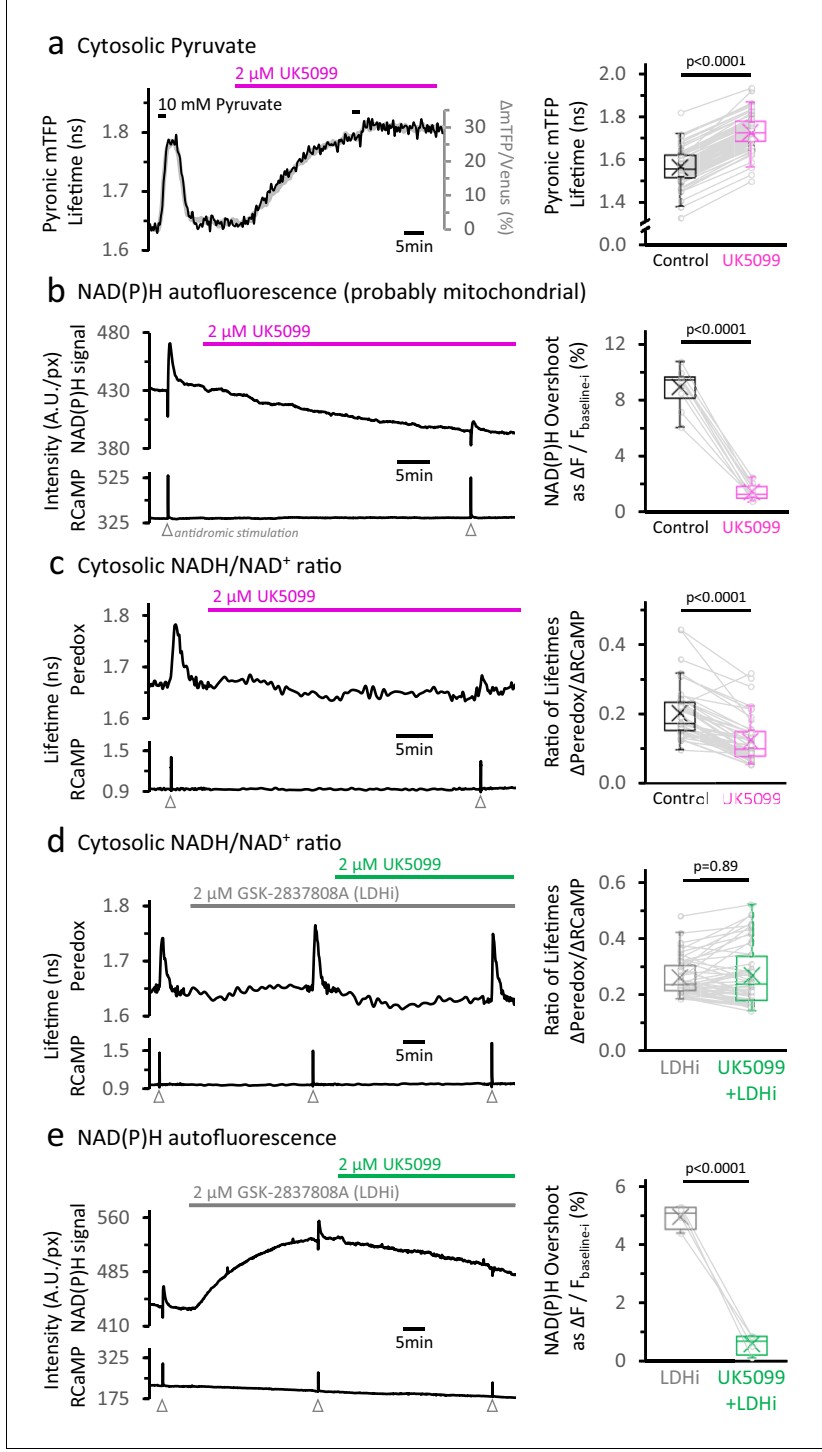

**Figure 2.** Inhibition of the mitochondrial pyruvate carrier reveals the independence of NADH transients in the cytosol and mitochondria. (a) *Left:* Representative traces of two different readouts of the Pyronic sensor, expressed in the cytosol of a dentate granule cell (DGC). The lifetime of the donor species mTFP (black line, left Y-axis) is overlapped to the change in the ratio of intensities between the donor and the acceptor species (Δ mTFP/Venus), expressed as the percent change over the initial mTFP/Venus value (gray line, right Y-axis). Both readouts transiently increase in response to a brief exposure to 10 mM pyruvate for 2 min in the bath solution. The application of 2 μM UK5099, an inhibitor of the mitochondrial pyruvate carrier, steadily increased the Pyronic signal to a plateau after 30–35 min. The bars indicate the times of application of both pyruvate and UK5099. *Right:* Quantification of Pyronic mTFP lifetimes before and after the treatment with UK5099. Data points obtained before

*Figure 2 continued on next page*

*Figure 2 continued*

and after the application of UK5099 are connected by lines. Box plots represent the 25–75% (Q2—Q3) interquartile range, and the whiskers expand to the lower (**Q1**) and upper (**Q4**) quartiles of the distribution (5–95%). The median of the distribution is represented by a horizontal line inside the box, and the mean is represented by a cross symbol (×). The data were compared using a paired Student's t's test ($N_{neurons}$ = 82, $N_{slices}$ = 7 and $N_{mice}$ = 4). (**b**) *Left:* Representative trace of the NAD(P)H autofluorescence signal (*top*), recorded from a population of DGCs in an acute hippocampal slice. These cells expressed the $Ca^{2+}$ sensor RCaMP1h, whose fluorescence was simultaneously monitored as a proxy for neuronal activity (*bottom*). A stimulating electrode was placed in the hilus of the dentate gyrus of the hippocampus, and a train of depolarizing pulses was delivered to the DGC axons (antidromic stimulation) before and after the treatment. Treating with UK5099 reduces the baseline and the responses induced by stimulation. *Right:* Quantification of the normalized amplitudes of the NAD(P)H signal overshoot, with or without UK5099. The baseline before each stimulation ($F_{baseline-i}$) was subtracted from the raw trace, and the difference between the baseline and the peak ($\Delta F = F_{peak} - F_{baseline-i}$) is presented as a percentage change over the baseline ($\Delta F/F_{baseline-i}$). The data were compared using a paired Student's t's test ($N_{slices}$ = 10 and $N_{mice}$ = 6). (**c**) *Left:* Representative trace of Peredox and RCaMP1h lifetimes simultaneously recorded in a DGC. The Peredox lifetime at baseline, and the metabolic transients in response to neuronal stimulation, were recorded before and after the application of UK5099. *Right:* The $NADH_{CYT}$ transient was decreased in the presence of UK5099. The Peredox lifetime change from the baseline to the peak of the transient was divided by the magnitude of the RCaMP1h transient ($\Delta$Peredox/$\Delta$RCaMP), as the metabolic responses are correlated with the $Ca^{2+}$ spikes (***Díaz-García et al., 2017***). The data were compared using a Wilcoxon matched pairs test ($N_{neurons}$ = 48, $N_{slices}$ = 7 and $N_{mice}$ = 6). (**d**) Representative trace of Peredox and RCaMP1h lifetimes simultaneously recorded in a DGC, with sequential application of the LDH inhibitor GSK-2837808A (LDHi) and UK5099. The data were compared using a Wilcoxon matched pairs test ($N_{neurons}$ = 52, $N_{slices}$ = 6 and $N_{mice}$ = 5). (**e**) Representative trace of the NAD(P)H autofluorescence signal and RCaMP1h fluorescence, simultaneously recorded from a population of DGCs in an acute hippocampal slice. Treating with 2 μM GSK-2837808A (LDH inhibitor) increases the baseline while preserving the waveform of the NAD(P)H responses to stimulation. The subsequent application of 2 μM UK5099, as in panel (**d**), diminished both phases of the NAD(P)H responses, especially the overshoot. The data were compared using a paired t-test ($N_{slices}$ = 4 and $N_{mice}$ = 3).

The online version of this article includes the following figure supplement(s) for figure 2:

**Figure supplement 1.** The rise in [pyruvate]$_{CYT}$ due to MPC inhibition almost saturates the Pyronic sensor.

**Figure supplement 2.** Inhibition of the mitochondrial pyruvate carrier slows down the TCA cycle and decreases $O_2$ consumption at baseline.

**Figure supplement 3.** After MPC inhibition, $O_2$ consumption and $FADH_2$ production are less affected than mitochondrial NADH production in acutely stimulated DGCs.

**Figure supplement 4.** MPC inhibition shortens the $NADH_{CYT}$ transient.

**Figure supplement 5.** LDH inhibition prevents the effect of UK5099 on the amplitude of the $NADH_{CYT}$ transient, but not on its time course.

**Figure supplement 6.** LDH inhibition elevates the NAD(P)H autofluorescence at baseline but does not restore the NAD(P)H overshoot after the treatment with UK5099.

---

Importantly, UK5099 provided a tool to abolish the mitochondrial NADH overshoot almost completely without compromising the health of the slices, as revealed by a low basal RCaMP signal and an unaltered $Ca^{2+}$ spike in response to stimulation (***Figure 2—figure supplement 2***). Using this manipulation, we tested the independence of the cytosolic NADH transient. We co-expressed the genetically encoded biosensors Peredox and RCaMP1h to monitor the cytosolic NADH/NAD$^+$ ratio and the $Ca^{2+}$ level, respectively, in individual DGCs. We measured the mean fluorescence lifetime (i.e. the time between photon absorption and photon emission), to obtain a direct readout of sensor occupancy (and analyte levels) that is independent of sensor expression level (***Yellen and Mongeon, 2015***). As previously reported (***Díaz-García et al., 2017***), antidromically stimulating DGCs induced a fast spike in RCaMP1h fluorescence lifetime and a slower increase in Peredox lifetime, consistent with a transient buildup of the NADH/NAD$^+$ ratio in the cytosol ($NADH_{CYT}$) (***Figure 2c*** Left, ***Figure 2—figure supplement 4***). This reflects an increase in glycolytic NADH production in the cytosol (***Díaz-García et al., 2017***). The cytosolic NADH/NAD$^+$ ratio, however, was not immune to MPC inhibition.

After MPC inhibition, there was reduction but not abolition of the $NADH_{CYT}$ transients: baseline Peredox lifetime decreased by 3.4 ± 1.6% (***Figure 2—figure supplement 4***) and the magnitude of the $NADH_{CYT}$ transients (expressed as the $\Delta$Peredox/$\Delta$RCaMP lifetime ratio) was attenuated by

almost 40% (*Figure 2c* Right), but far less than the ~7-fold reduction in the mitochondrial overshoot transient. Are the reduced transients evidence that cytosolic and mitochondrial NADH responses to stimulation are inextricably connected? Not necessarily: an alternative explanation is that by accumulating pyruvate in the cytosol, MPC inhibition indirectly drives increased NADH re-oxidation via the LDH reaction. Indeed, in the continuous presence of an LDH inhibitor to oppose the pyruvate-driven re-oxidation, further application of UK5099 did not change the magnitude of the $NADH_{CYT}$ transient in response to stimulation (*Figure 2d*, *Figure 2—figure supplement 5*). This result indicates that the stimulation-induced acceleration of neuronal glycolysis, as detected by NADH production in the cytosol, is not affected by impaired NADH production in the mitochondria. MPC inhibition still abolished the mitochondrial NAD(P)H transients in the presence of the LDH inhibitor, by about 8-fold (similar to the effect without LDH inhibition; *Figure 2e*, *Figure 2—figure supplement 6*). Taken together, these results show that NADH transients in the cytosol develop independently from transients in the mitochondria.

## Calcium entry into mitochondria is required for strong activation of TCA metabolism

The NAD(P)H autofluorescence signals and the $NADH_{CYT}$ transients, which report on different biochemical reactions, are both rapidly triggered by neuronal stimulation. Calcium ions, which enter the cytosol via voltage-gated $Ca^{2+}$ channels during action potentials, could facilitate the concerted glycolytic and mitochondrial responses. The rise in $[Ca^{2+}]_{CYT}$ can propagate to the mitochondria, where $Ca^{2+}$ transiently dissipates the inner mitochondrial membrane (*Duchen, 1992*) and activates several dehydrogenases in the TCA cycle (*McCormack et al., 1990*).

Many studies have explored the effects of $Ca^{2+}$ on the different phases of the NAD(P)H response, yielding contradictory results. Experiments using $Ca^{2+}$-free extracellular solutions have been particularly hard to reconcile, with reports showing either a complete elimination of both phases of the NAD(P)H transient (*Duchen, 1992*) or the preservation of the entire signal, depending on the stimulation paradigm (e.g. with kainate, *Shuttleworth et al., 2003*). We hypothesized that $Ca^{2+}$ can contribute to both the dip and the overshoot of the NAD(P)H signal, although not necessarily to the same extent.

To test this hypothesis, we sought to disrupt the expression of the mitochondrial $Ca^{2+}$ uniporter (MCU), an ion channel located at the inner mitochondrial membrane that constitutes the dominant path for mitochondrial $Ca^{2+}$ uptake (*Kirichok et al., 2004*; *Baughman et al., 2011*; *De Stefani et al., 2011*). This approach has been proven effective in diminishing $Ca^{2+}$ influx into the mitochondria—but not the cytosolic $Ca^{2+}$ spike—in axons (*Ashrafi et al., 2020*) and cardiomyocytes (*Kwong et al., 2015*).

We optimized the knockdown of MCU (MCU-KD) by expressing Cre recombinase under the Dock10 promoter (*Kohara et al., 2014*), in DGCs of adult hemizygous $Mcu^{fl/\Delta}$ mice (derived from $Mcu^{fl/fl}$ mice; *Kwong et al., 2015*), which resulted in a strong reduction of the mitochondrial $Ca^{2+}$ transients monitored with a mitochondrially targeted sensor mitoRCaMP1h, as well as lower resting mitochondrial $[Ca^{2+}]$ (*Figure 3b*, *Figure 3—figure supplements 1* and *2*). Nevertheless, we still observed prominent peaks immediately after stimulating the slices, in a manner that resembles the rapid $Ca^{2+}$ transient detected with a cytosolic RCaMP sensor. This rapid initial phase in the mitoRCaMP signal may reflect either a fraction of mistargeted sensor that is not fully translocated into the mitochondrial matrix, a rapid mode of $Ca^{2+}$ uptake into the mitochondria (*Sparagna et al., 1995*), or fast dissociation/association of calcium phosphate deposits upon transient matrix acidification (*Hernansanz-Agustín et al., 2020*).

The smaller mitochondrial $Ca^{2+}$ influx in MCU-KD neurons leads to smaller mitochondrial NADH transients. Indeed, the prominent NAD(P)H overshoots observed in DGCs from control $Mcu^{fl/\Delta}$ mice (~2.1—4.8% for 25—200 pulse stimulations) were strongly attenuated in MCU-KD neurons from $Mcu^{fl/\Delta}$ Dock10Cre mice (*Figure 3a*). The NAD(P)H overshoot was almost abolished in the 25-pulse stimulation, and strongly reduced but not quite abolished when stimulating with 200 pulses (*Figure 3b*), perhaps reflecting $Ca^{2+}$ entry into the mitochondria via remnant MCU. The strong effect of $Ca^{2+}$ deprivation on the NAD(P)H overshoot together with a delayed recovery from the dip (*Figure 3—figure supplement 3*) are consistent with a major role for $[Ca^{2+}]_{MITO}$ in activating dehydrogenases in the TCA cycle to resupply mitochondrial NADH after neuronal stimulation. Interestingly,

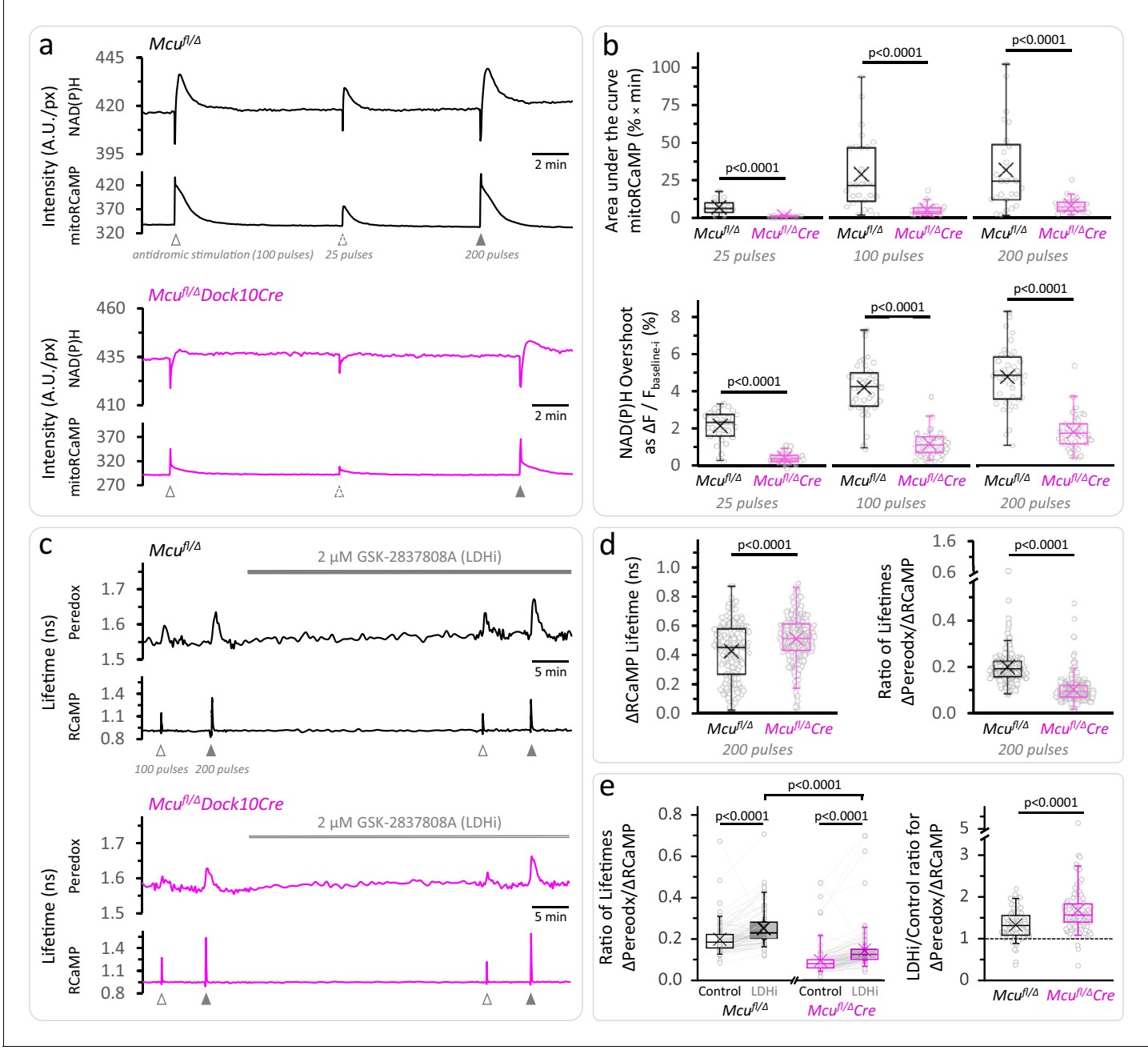

**Figure 3.** Calcium entrance via MCU is essential for the mitochondrial NAD(P)H overshoot. (a) Representative traces of the NAD(P)H autofluorescence signals and the mitoRCaMP1h transients in response to different trains of depolarizing pulses. The data correspond to simultaneously recorded signals from a population of DGCs in acute hippocampal slices. (b) *Top*: Overall Ca$^{2+}$ entry into the mitochondria was estimated using the area under the curve of the mitoRCaMP signal, a measure that is resistant to the very brief transient that remains in the MCU-KD experiments. It was compared between $Mcu^{fl/\Delta}$ (*25 pulses*: $N_{slices}$ = 17, $N_{mice}$ = 10; *100 pulses*: $N_{slices}$ = 27, $N_{mice}$ = 12; *200 pulses*: $N_{slices}$ = 26, $N_{mice}$ = 12) and $Mcu^{fl/\Delta}Dock10Cre$ mice (abbreviated $Mcu^{fl/\Delta}Cre$ in the figure, *25 pulses*: $N_{slices}$ = 21, $N_{mice}$ = 9; *100 pulses*: $N_{slices}$ = 31, $N_{mice}$ = 13; *200 pulses*: $N_{slices}$ = 30, $N_{mice}$ = 12). Data points represent individual slices. Only recordings with a peak amplitude ≥1% over the baseline were included for analysis. The area under the curve was calculated using the trapezoidal rule as the product of the signal amplitude (in %) and the acquisition interval (in min), for 2 min after the beginning of the stimulation. All comparisons were performed using a Mann-Whitney test. *Bottom*: The NAD(P)H overshoot was diminished in Cre-expressing DGCs of $Mcu^{fl/\Delta}$ mice. The datasets used for comparisons included slices that did not expressed mitoRCaMP in $Mcu^{fl/\Delta}$ (*25 pulses*: $N_{slices}$ = 31, $N_{mice}$ = 17; *100 pulses*: $N_{slices}$ = 41, $N_{mice}$ = 19; and *200 pulses*: $N_{slices}$ = 40, $N_{mice}$ = 19) and $Mcu^{fl/\Delta}Dock10Cre$ (*25 pulses*: $N_{slices}$ = 39, $N_{mice}$ = 19; *100 pulses*: $N_{slices}$ = 49, $N_{mice}$ = 23; and *200 pulses*: $N_{slices}$ = 48, $N_{mice}$ = 22). All comparisons were performed using an unpaired t-test with a Welch's correction. (c) Representative traces of Peredox and RCaMP1h lifetimes simultaneously recorded in DGCs of both genotypes. The typical stimulation protocol of 100 pulses elicited a smaller response (both in RCaMP and Peredox) in adult animals compared to juvenile mice. Increasing the number of

*Figure 3 continued on next page*

*Figure 3 continued*

pulses to 200 improved the detection of the NADH$_{CYT}$ transients. (**d**) *Left:* Comparison of the ΔRCaMP lifetime transient between *Mcu*$^{fl/Δ}$ mice (N$_{neurons}$ = 197, N$_{slices}$ = 21 and N$_{mice}$ = 9) and *Mcu*$^{fl/Δ}$*Dock10Cre* mice (N$_{neurons}$ = 275, N$_{slices}$ = 27 and N$_{mice}$ = 11). The range of the data is similar between genotypes, although the mean Ca$^{2+}$ transient is higher in *Mcu*$^{fl/Δ}$*Dock10Cre* mice. It is unlikely that this reflects an impaired clearance of cytosolic Ca$^{2+}$ by mitochondria, since the transients in MCU-lacking DGCs were similar to neurons from *Mcu*$^{fl/fl}$ mice (**Figure 3—figure supplement 4**). *Right:* The NADH$_{CYT}$ transient was decreased in DGCs lacking MCU, as revealed by the comparison of the ratio of the Peredox lifetime change, divided by the magnitude of the RCaMP1h transient (to normalize the metabolic responses to the elevation in [Ca$^{2+}$]$_{CYT}$). (**e**) *Left:* Pyruvate accumulation is partially responsible for the diminished NADH$_{CYT}$ transients in *Mcu*$^{fl/Δ}$*Dock10Cre* mice, but there might be other mechanisms contributing to this difference. In a subset of slices that were treated with the LDH inhibitor, the ΔPeredox/ΔRCaMP ratio increased in both groups but it did not equalize the transients in the two genotypes; DGCs from *Mcu*$^{fl/Δ}$*Dock10Cre* mice (N$_{neurons}$ = 113, N$_{slices}$ = 12 and N$_{mice}$ = 6) remained about ~57% of hemizygote *Mcu*$^{fl/Δ}$ mice (N$_{neurons}$ = 103, N$_{slices}$ = 12 and N$_{mice}$ = 6). These comparisons were performed using a paired Wilcoxon test. *Right*: Consistent with cytosolic pyruvate accumulation in the MCU-KD, LDH inhibition produced a larger increase of the ΔPeredox/ΔRCaMP ratio in the MCU-KD. Mann-Whitney test was used for all comparisons except where stated otherwise.

The online version of this article includes the following figure supplement(s) for figure 3:

**Figure supplement 1.** Design of the mitochondrially targeted RCaMP1h.

**Figure supplement 2.** Mitochondrial Ca$^{2+}$ levels at rest and during stimulation are lower in DGCs from *Mcu*$^{fl/Δ}$ *Dock10Cre* mice, compared to *Mcu*$^{fl/Δ}$ control mice.

**Figure supplement 3.** The initial dip of the NAD(P)H signal, the overall FAD$^+$ signal, and the O$_2$ dip in response to stimulation, are less affected than the slow NAD(P)H overshoot in DGCs from *Mcu*$^{fl/Δ}$ *Dock10Cre* mice.

**Figure supplement 4.** Loss of one *Mcu* allele slightly reduces the NADH$_{CYT}$ transient.

**Figure supplement 5.** Hemizygous mice for MCU show mild impairments in the TCA cycle with respect to *Mcu*$^{fl/fl}$ controls.

the reductive phase of the FAD$^+$ signal, although attenuated, was more resistant to MCU depletion than the NAD(P)H overshoot (**Figure 3—figure supplement 3**).

On the other hand, the magnitude of the initial NAD(P)H dip remained unaffected in neurons lacking MCU, except for a small ~19% decrease when stimulated with 25 pulses (**Figure 3—figure supplement 3**). This suggests that Ca$^{2+}$ influx through MCU is only a minor contributor to the dissipation of the mitochondrial membrane potential (Δψ$_m$), which promotes the oxidation of NADH by Complex I (seen as the dip in autofluorescence). Consistent with the changes observed in the NAD(P)H dip, the amplitude of the transient decrease in the tissue [O$_2$] was only diminished by MCU-KD when the slices were stimulated with 25 pulses, while the O$_2$ decrease was delayed but not diminished in amplitude when using a higher number of pulses (**Figure 3—figure supplement 3**).

The diminished mitochondrial Ca$^{2+}$ transient in the MCU-KD also produced some decrease in the magnitude of NADH$_{CYT}$ transient (**Figure 3c,d**). This decrease was ameliorated by LDH inhibition (as seen previously for the case of MPC inhibition), but the MCU-KD transient was still only ~57% of the control hemizygote (**Figure 3e**, **Figure 3—figure supplement 4**), indicating that NADH re-oxidation by LDH cannot be the sole cause for this difference. A potential explanation for the remaining effect of MCU-KD on the NADH$_{CYT}$ transient could be that reoxidation of cytosolic NADH by the MAS runs faster in MCU-KD neurons during stimulation, because it is not inhibited by the usual increases in [Ca$^{2+}$]$_{MITO}$ (**Bak et al., 2012**; **Figure 1—figure supplement 1**).

It is also possible that there is increased expression of one or more components of the MAS, or the glycerol-phosphate shuttle, to compensate for the lower TCA cycle activity due to chronic MCU inhibition. In addition, the ability of these mitochondrial shuttles in clearing NADH from the cytosol should be more prominent during stimulation, since their activities increase in response to rises in cytosolic Ca$^{2+}$ (**Rutter et al., 1992**; **Pardo et al., 2006**; reviewed by **McKenna et al., 2006**; **Figure 1—figure supplement 1**).

In any case, our experiments show that despite the strong effect of knocking out *Mcu* on the overshoot of the mitochondrial NAD(P)H autofluorescence signal, robust NADH increases were consistently elicited in the cytosol of stimulated neurons, especially when NADH re-oxidation through LDH was prevented.

## Calcium elevation in the cytosol is a major contributor to the NADH$_{CYT}$ transients

The amplitude of the NADH$_{CYT}$ responses, reflecting a temporary increase in aerobic glycolysis, correlates very closely with the cytosolic Ca$^{2+}$ transient elicited during stimulation (**Díaz-García et al.**,

*2017*). However, the mechanism linking these two events is still unknown. To test the role of $Ca^{2+}$ in the cytosolic $NADH_{CYT}$ response, we used three orthogonal approaches to diminish the rise in $[Ca^{2+}]_{CYT}$ upon stimulation while preserving other ionic fluxes. First, we blocked L-type $Ca^{2+}$ channels with 3 μM isradipine (reviewed in *Catterall et al., 2005*; *Striessnig et al., 2015*), which reduced the cytosolic $Ca^{2+}$ transient in response to stimulation by 51 ± 10% (*Figure 4—figure supplement 1*), and this reduced the $NADH_{CYT}$ transient by 36 ± 13% (*Figure 4a*). We then added 20 μM of the non-selective $Ca^{2+}$ channel inhibitor $CdCl_2$, which further reduced the $Ca^{2+}$ transient by 81 ± 7% from the original value while the magnitude of the $NADH_{CYT}$ transients dropped 71 ± 11% (*Figure 4a*).

Second, we applied 100 μM of the cell-permeable $Ca^{2+}$ chelator EGTA-AM. This manipulation preserves the $Ca^{2+}$ influx upon stimulation, but prevents the rise in the intracellular concentration of free $Ca^{2+}$ ions. The effect on the $NADH_{CYT}$ transient was similar to the previous results with inhibition of $Ca^{2+}$ influx: after ~1 hr in EGTA-AM, the stimulus-induced change in the $Ca^{2+}$ signal was reduced by 87 ± 6% and the $NADH_{CYT}$ transient was reduced by 73 ± 11% (*Figure 4b*). (Because $Ca^{2+}$ buffering seemed less effective in the dendrites compared to the somata, the average RCaMP responses in the slice were kept small, ≤0.6 ns, to prevent the less attenuated $Ca^{2+}$ spike in the dendrites from triggering an $NADH_{CYT}$ response that could be detected in the soma. A control experiment with 0.1% DMSO (*Figure 3—figure supplement 1*) showed no change in the $NADH_{CYT}$ transients during the typical duration of our experiments.)

Finally, we perfused the slices with a nominally $Ca^{2+}$-free ACSF, by replacing all extracellular $CaCl_2$ with a concentration of $MgCl_2$ chosen to match the charge screening effects on the plasma membrane (*Hille et al., 1975*), to avoid changes in the effective voltage-dependence of ion channels. Additionally, 1 mM EGTA was included to ensure the chelation of any residual $Ca^{2+}$ in the extracellular space within the slice. As expected, the transient RCaMP signal almost disappeared in the absence of extracellular $Ca^{2+}$ (a 91 ± 5% reduction) and, just as with previous manipulations, the $NADH_{CYT}$ dropped by 73 ± 11% from the initial responses (*Figure 4c*). A variation of this experiment using a nominal zero $Ca^{2+}$ solution without EGTA was similarly effective in decreasing the $NADH_{CYT}$ transient (*Figure 4—figure supplement 2*).

Once again we tested the role of NADH re-oxidation via the LDH reaction, since depriving the mitochondria of a $Ca^{2+}$ transient would diminish their pyruvate consumption. We repeated each of the $Ca^{2+}$ manipulations in the presence of 2 μM of the LDH inhibitor GSK-2837808; they still decreased the $NADH_{CYT}$ transients by more than 50% but not to zero (*Figure 4—figure supplement 1*).

Overall, these results indicate that $Ca^{2+}$ elevation resulting from the activation of several types of $Ca^{2+}$ channels (including the L-type), is a major contributor to the glycolytic response in stimulated neurons but it is not strictly required.

Calcium is known to modulate many signaling pathways that can act as a feedforward mechanism to promote energy production in anticipation of another episode of neuronal activity. Could neuronal stimulation trigger glycolysis via one of these pathways? One ubiquitous signaling pathway in mammalian cells is the $Ca^{2+}$/calmodulin axis, which has many downstream targets that promote glucose utilization (*Marsin et al., 2000*; *Marinho-Carvalho et al., 2009*; *Schmitz et al., 2013*; *Singh et al., 2004*; *Xie et al., 2014*; *Kim et al., 2016*). If this is a major signal in triggering the $NADH_{CYT}$ transients, we would expect them to decrease by inhibiting the $Ca^{2+}$/calmodulin complex with E6-berbamine or calmidazolium. However, these drugs caused the opposite effect, increasing the $NADH_{CYT}$ transients relative to the $Ca^{2+}$ spike (*Figure 5a,b*). Both drugs increased the baseline for RCaMP by ~3% but did not hamper the $Ca^{2+}$ responses to stimulation (*Figure 5—figure supplement 1a,b*). These results suggest that $Ca^{2+}$ does not activate the fast glycolytic response via $Ca^{2+}$/calmodulin dependent signaling.

We also tested if AMPK, a protein kinase that senses the cellular energy status through AMP levels and is also modulated by $Ca^{2+}$ (*Hawley et al., 2005*; *Woods et al., 2005*; reviewed by *Herzig and Shaw, 2018*), could be responsible for the enhanced glycolysis upon stimulation. The application of the inhibitor dorsomorphin (also known as Compound C) rapidly increased both RCaMP and Peredox baselines by ~8% and 3%, respectively. Dorsomorphin decreased the $Ca^{2+}$ transient by 18 ± 10% (*Figure 5—figure supplement 1c*) but a commensurate decrease in $NADH_{CYT}$ transients was also observed, keeping the ΔPeredox/ΔRCaMP ratio unaffected (*Figure 5c*). Furthermore, the $NADH_{CYT}$ transients became briefer in the presence of dorsomorphin due to a faster time

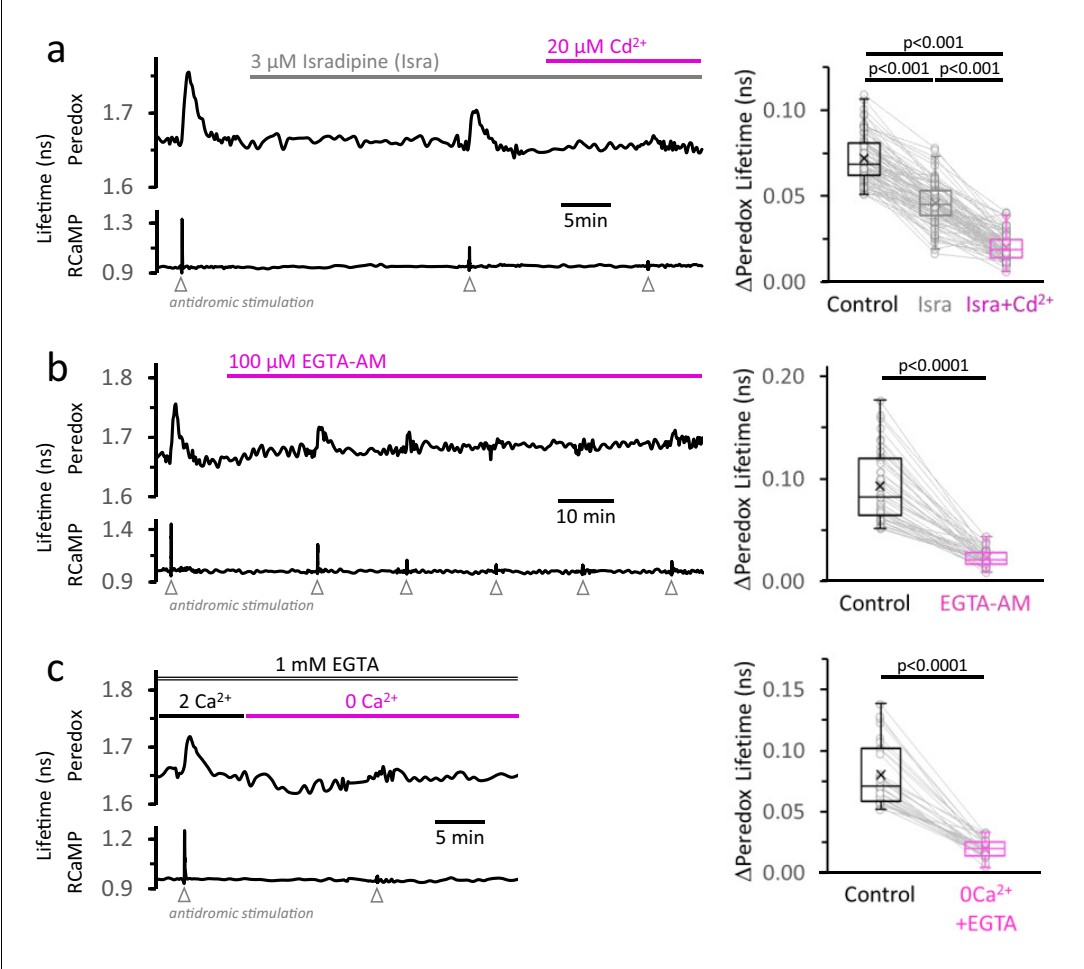

**Figure 4.** The rise in $[Ca^{2+}]_{CYT}$, mainly caused by the activity of high-voltage-activated $Ca^{2+}$ channels, makes a major contribution to the $NADH_{CYT}$ transients in response to stimulation. (a) *Left:* Representative trace from a DGC expressing Peredox and RCaMP1h. The slice was superfused for ~20 min with the L-type $Ca^{2+}$ channel inhibitor isradipine (Isra, 3 µM), and then stimulated. In the continuous presence of isradipine, 20 µM of $CdCl_2$ ($Cd^{2+}$, a non-selective blocker of voltage-activated $Ca^{2+}$ channels) was added to the ACSF. Inhibition of $Ca^{2+}$ influx was evident from the progressive reduction of the stimulus associated RCaMP1h spike. *Right:* The amplitude of the metabolic responses to stimulation (Peredox lifetime change) mirrored the decrease in the RCaMP spikes (*Figure 4—figure supplement 1a*). The data were compared using a non-parametric repeated measures ANOVA (Friedman test) with a Dunn post-test ($N_{neurons}$ = 86, $N_{slices}$ = 11 and $N_{mice}$ = 6). For all panels, only neurons showing an initial ΔPeredox lifetime response $\geq$0.05 ns were included for analysis. (b) *Left:* Representative trace of a DGC superfused with EGTA-AM (100 µM), a cell-permeable $Ca^{2+}$ chelator. As expected, the stimulus-induced RCaMP transients gradually diminished over time, typically stabilizing after ~1 hr of treatment. *Right:* $NADH_{CYT}$ transients are strongly attenuated after effective $Ca^{2+}$ buffering by EGTA-AM (*Figure 4—figure supplement 1b*). The data were compared using a Wilcoxon matched pairs test ($N_{neurons}$ = 45, $N_{slices}$ = 5 and $N_{mice}$ = 5). (c) *Left:* Representative trace for the effect of $Ca^{2+}$ removal from the bath solution on the metabolic transients in the cytosol. The cell-impermeant $Ca^{2+}$ chelator EGTA (1 mM) was added to the ACSF to reinforce $Ca^{2+}$ removal after switching to a nominal $0Ca^{2+}$ solution. A modified control ACSF also contained 1 mM EGTA and an adjusted total $[Ca^{2+}]$ resulting in a free concentration of 2 mM, as in any other control experiment. Effective $Ca^{2+}$ removal was confirmed by the absence of a RCaMP1h spike upon stimulation. *Right:* The $NADH_{CYT}$ transients were diminished in a $Ca^{2+}$-deprived ACSF. The data were compared using a Wilcoxon matched pairs test ($N_{neurons}$ = 31, $N_{slices}$ = 7 and $N_{mice}$ = 6).

The online version of this article includes the following figure supplement(s) for figure 4:

**Figure supplement 1.** A major $Ca^{2+}$-dependent component of the $NADH_{CYT}$ also occurs under LDH inhibition.

**Figure supplement 2.** Calcium removal from the ACSF, without addition of EGTA, is also effective in decreasing the $NADH_{CYT}$ transients.

**Figure supplement 3.** DMSO control for the EGTA-AM experiments.

to peak and recovery to baseline (*Figure 5—figure supplement 1c*). Taken together, these results show that AMPK is not necessary to increase neuronal glycolysis upon stimulation, although it can modulate the duration of this metabolic response.

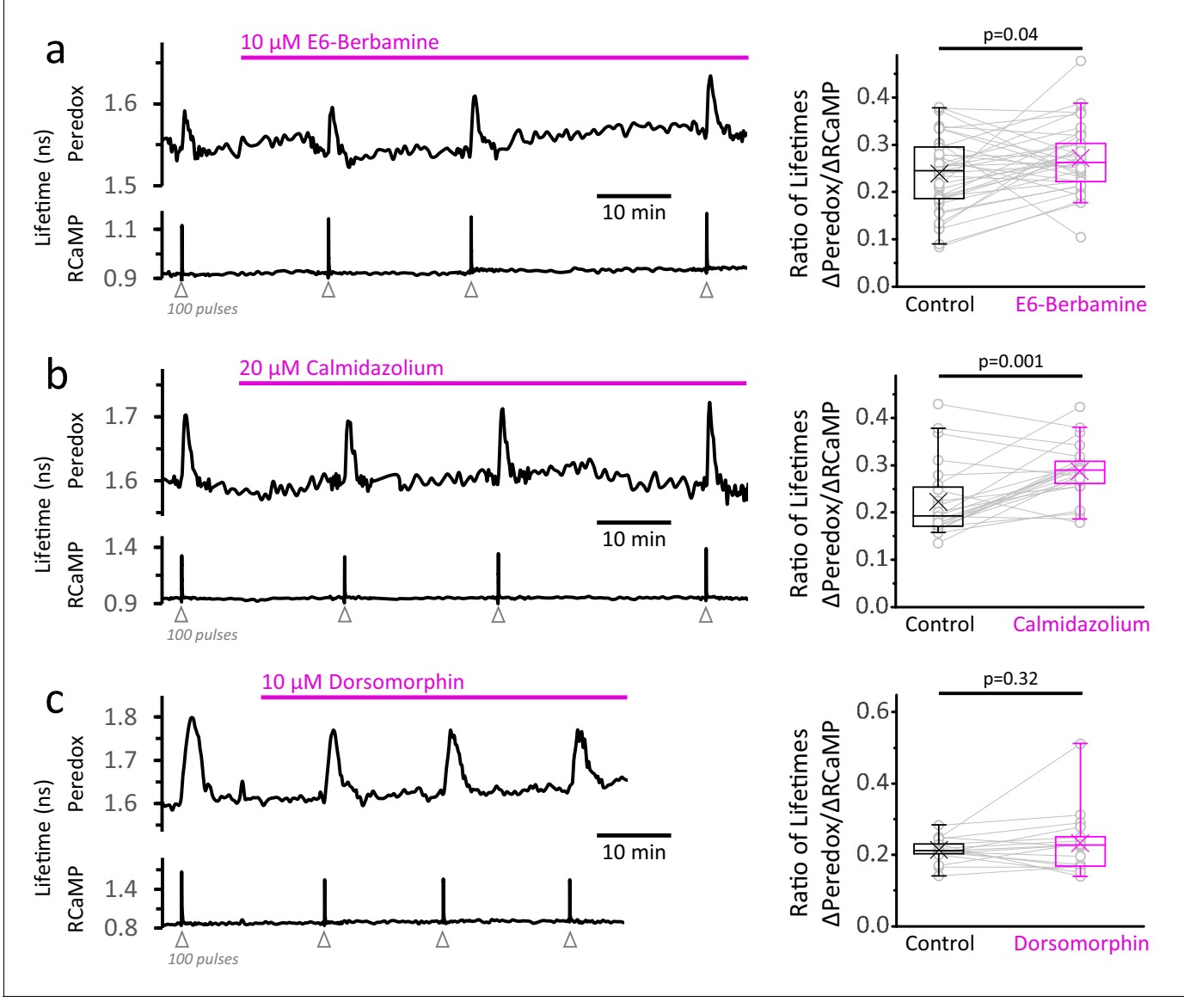

**Figure 5.** Inhibition of the $Ca^{2+}$-calmodulin complex or of AMPK does not abolish the $NADH_{CYT}$ transients in response to stimulation. (**a**) *Left:* Representative trace of a DGC treated with E6-berbamine (10 µM), an inhibitor of the $Ca^{2+}$-calmodulin signaling pathway. *Right:* The magnitude of the metabolic response was expressed as the change in Peredox lifetime change divided by the change in RCaMP1h lifetime in response to stimulation ($\Delta$Peredox/$\Delta$RCaMP). E6-berbamine marginally elevated the $NADH_{CYT}$ transients. The data were compared using a paired t-test ($N_{neurons}$ = 37, $N_{slices}$ = 4 and $N_{mice}$ = 4). For all panels, the effect of the drugs on the metabolic transients were monitored for at least 30 min. (**b**) *Left:* Representative trace of a DGC treated with calmidazolium (20 µM), another inhibitor of the $Ca^{2+}$-calmodulin signaling pathway. *Right:* The effect of calmidazolium on the $\Delta$Peredox/$\Delta$RCaMP ratio was similar to the previous $Ca^{2+}$-calmodulin complex inhibitor. The data were compared using a Wilcoxon matched pairs test ($N_{neurons}$ = 24, $N_{slices}$ = 3 and $N_{mice}$ = 3). (**c**) *Left:* Representative trace of a DGC before and after the inhibition of the AMPK pathway using dorsomorphin (10 µM). Since the drug decreases the $Ca^{2+}$ transients (***Figure 5—figure supplement 1c***), the intensity of the stimulus was adjusted to elicit strong responses, and the number of depolarizing pulses was sometimes increased to 150 (before and after treatment) to ensure effective stimulation of the soma throughout the experiment. *Right:* In the presence of dorsomorphin, the $\Delta$Peredox/$\Delta$RCaMP ratio remained unaltered. The data were compared using a paired t-test ($N_{neurons}$ = 17, $N_{slices}$ = 4 and $N_{mice}$ = 3).

The online version of this article includes the following figure supplement(s) for figure 5:

**Figure supplement 1.** $Ca^{2+}$/CaM and AMPK signaling modulates the cytosolic $[Ca^{2+}]$ and $NADH/NAD^+$ ratio at baseline, and the time course of the $NADH_{CYT}$ transient in response to stimulation.

## Energy demand from Na$^+$ or Ca$^{2+}$ pumping triggers aerobic glycolysis

While the cytosolic NADH/NAD$^+$ ratio is sensitive to Ca$^{2+}$-dependent signaling pathways, inhibition of these pathways did not prevent the NADH$_{CYT}$ transient in response to stimulation. More importantly, activity-induced increases in [Ca$^{2+}$]$_{CYT}$ may not be *necessary* for the metabolic transient, although this increase apparently accounts for a surprisingly large part of the metabolic transient. It may be that the fast glycolytic response is simply reactive to the energy demand resulting from Ca$^{2+}$ extrusion, and that this is a large fraction of the total energy demand produced by activity. Restoring [Ca$^{2+}$]$_{CYT}$ to pre-stimulation levels requires the activity of ion pumps that move Ca$^{2+}$ out of the cytosol at the expense of ATP hydrolysis (*Figure 1—figure supplement 2*). The resulting ADP, either by itself or in combination with AMP (which can be produced from ADP via the adenylate kinase reaction) could then trigger neuronal glycolysis. Other ions that enter during stimulation such as Na$^+$ should also contribute to energy demand, though pumping of Na$^+$ ions requires less ATP per ion. In the somata of central neurons with brief action potentials, the contribution of Na$^+$ might still be expected to exceed that of Ca$^{2+}$, but the Ca$^{2+}$ contribution can be underestimated due to countervailing Ca$^{2+}$-activated K$^+$ currents (*Bean, 2007*; *Brenner et al., 2005*).

If the main consequence of Ca$^{2+}$ entry during stimulation is to increase energy demand, we should be able to restore the NADH$_{CYT}$ transients by increasing energy demand independent of Ca$^{2+}$. We tested this hypothesis by boosting Na$^+$ influx, which would lead to a greater ATP hydrolysis by the Na$^+$/K$^+$ ATPases.

We started by preventing the cytosolic Ca$^{2+}$ elevation in DGCs with a nominally Ca$^{2+}$-free external solution (supplemented with EGTA), in the continuous presence of the LDH inhibitor GSK-2837808A to maximize the glycolytic NADH$_{CYT}$ transients. Then we applied α-pompilidotoxin, a toxin that slows down voltage-gated Na$^+$ channel inactivation (*Konno et al., 1998*; *Schiavon et al., 2010*) and should increase Na$^+$ influx during action potentials, increasing energy demand upon stimulation. Indeed, in the absence of a Ca$^{2+}$ spike, the application of 10 μM α-pompilidotoxin induced a large ~3.8-fold increase in the magnitude of the Peredox transient, recovering ~81% of the initial response in regular ACSF (*Figure 6a*), although with slightly faster kinetics (*Figure 6a*, *Figure 6—figure supplement 1*). This is consistent with the hypothesis of a metabolic response that is reactive to energy demand, and constitutes direct evidence that although Ca$^{2+}$ largely determines the magnitude of the cytosolic transients in physiological conditions, it is not strictly necessary for triggering these responses. This is fundamentally different from the overshoot of NAD(P)H transients, which is not recovered by a similar manipulation, even if a higher concentration of α-pompilidotoxin was applied (*Figure 6—figure supplement 2*).

As a corollary of this experiment, the increase in glycolysis should be prevented by diminishing the ADP surge associated with ion pumping. We sought evidence for this by inhibiting the Na$^+$/K$^+$ pumps with strophanthidin. In virtue of its relatively high octanol/water partition coefficient (*Dzimiri et al., 1987*), strophanthidin provides extra convenience over other cardiac glycosides in that it blocks the activity of the Na$^+$/K$^+$ pumps regardless of the subcellular localization of the pumps (*Galva et al., 2012*). Indeed, co-application of 10 μM strophanthidin reversed the increases in NADH$_{CYT}$ transients produced by α-pompilidotoxin (3 μM), and in some cases decreased the transients even below pre-α-pompilidotoxin levels (*Figure 6c*). These results confirmed that ATP hydrolysis by ion pumping is the main factor that triggers neuronal glycolysis in response to stimulation.

## Discussion

### Mitochondrial Ca$^{2+}$ uptake is required for increasing mitochondrial NADH upon neuronal stimulation

Calcium orchestrates increases in both glycolysis and the TCA cycle in response to acute neuronal stimulation. Changes in these metabolic pathways reflect the compartmentalization of Ca$^{2+}$ dynamics: while Ca$^{2+}$ influx through voltage-gated ion channels promotes glycolysis in the cytosol via the energy associated to the restoration of ionic gradients at the plasma membrane, further Ca$^{2+}$ uptake into the mitochondria is required for increasing NADH production in this organelle. Moreover, cytosolic Ca$^{2+}$ can activate the aspartate-glutamate mitochondrial carrier one and thus the malate-aspartate shuttle, which may act as a source of NADH for the mitochondrial matrix when intramitochondrial Ca$^{2+}$ is low (*Pardo et al., 2006*).

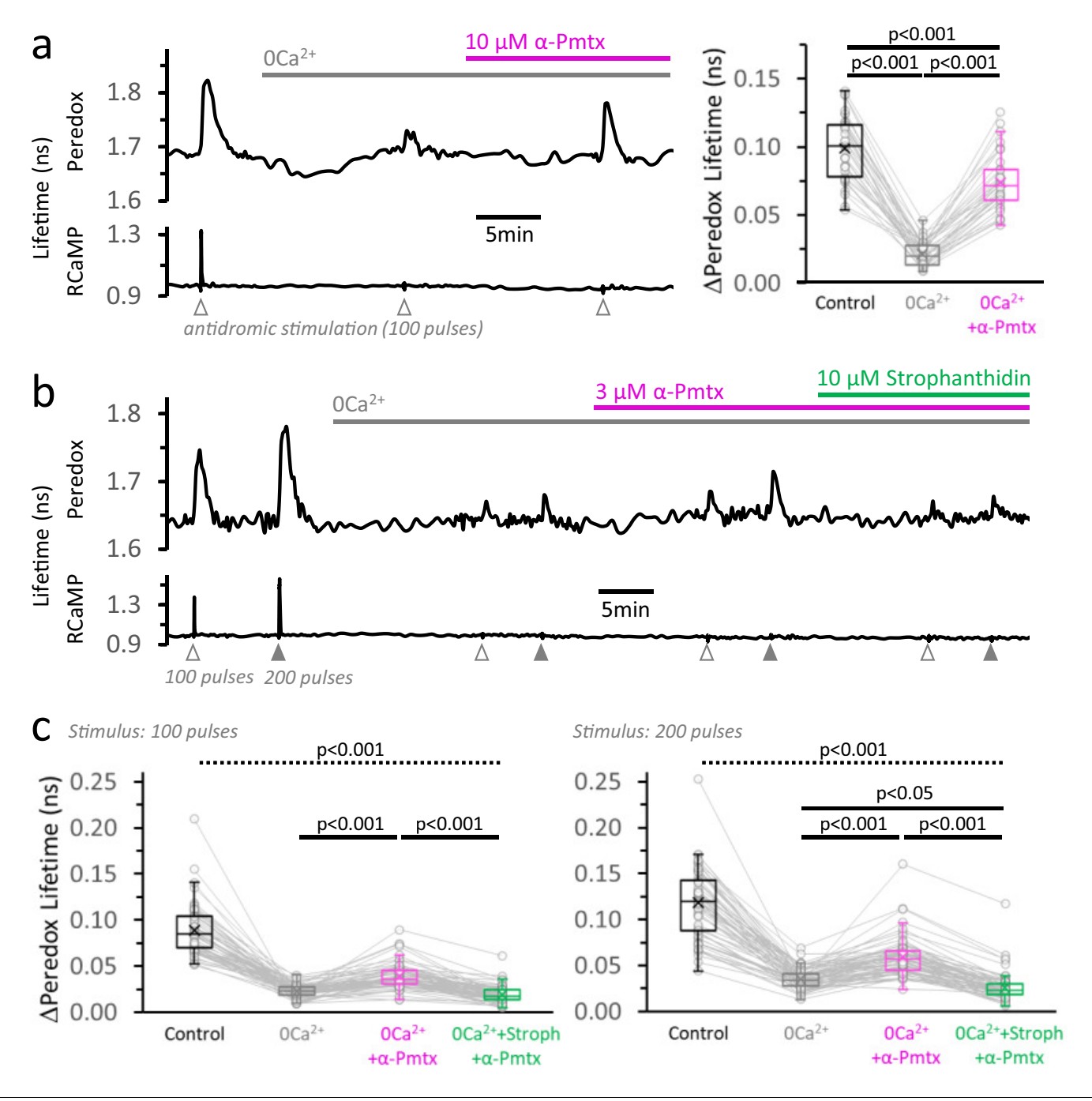

**Figure 6.** Neuronal stimulation triggers glycolysis in response to energy demand from ion pumping. (a) *Left:* Representative trace of Peredox and RCaMP1h lifetimes simultaneously recorded in a DGC from an acute hippocampal slice. The slice was superfused with 2 μM GSK-2837808A for at least 30 min before the experiment, and the LDH inhibitor was kept in the ACSF during the experiment. The ACSF also contained 1 mM EGTA to reinforce $Ca^{2+}$ removal in the nominal $0Ca^{2+}$ condition (but $[Ca^{2+}]$ in the control ACSF was accordingly adjusted to a free concentration of 2 mM, as in any other experiment). Effective $Ca^{2+}$ removal was confirmed by the absence of a RCaMP1h spike upon stimulation. The Peredox lifetime at baseline, and the metabolic transients in response to neuronal stimulation, were recorded after substituting the bath solution with a $0Ca^{2+}$ ACSF (to obtain $Na^+$-only $NADH_{CYT}$ responses), and the further application of 10 μM α-pompilidotoxin (α-Pmtx, a toxin that prevents voltage-gated $Na^+$ channel inactivation). *Right:* The $Na^+$-only $NADH_{CYT}$ transient was increased in the presence of α-pompilidotoxin. The Peredox lifetime change in response to stimulation was diminished in the absence of $Ca^{2+}$ but bounced back to higher amplitudes by increasing $Na^+$ influx. The data were compared using a repeated measures ANOVA with a Student-Newman-Keuls post-test ($N_{neurons}$ = 35, $N_{slices}$ = 5 and $N_{mice}$ = 3). For all panels, only neurons showing an initial ΔPeredox lifetime response ≥0.05 ns were included for analysis. (b) Representative trace of Peredox and RCaMP1h lifetimes in a DGC stimulated with

*Figure 6 continued on next page*

*Figure 6 continued*

trains of 100 and 200 electrical pulses. The two-stimulation protocol was also performed in $0Ca^{2+}$ ACSF before and after the application of 3 μM α-pompilidotoxin. The latter condition was followed by the application of the $Na^+/K^+$ ATPase inhibitor strophanthidin. As in (a), the slices were exposed to the LDH inhibitor GSK-2837808A from 30 min prior, until the end of the experiment. Likewise, all the solutions contained 1 mM EGTA. (c) Comparison of the Peredox lifetime changes in response to both stimulation paradigms (100 or 200 pulses) among the conditions in (b). The $NADH_{CYT}$ transients in control condition was different from the other conditions (the discontinuous line for the associated p-value applies to all comparisons). The $Na^+$-only $NADH_{CYT}$ responses recorded in $0Ca^{2+}$ ACSF were increased slightly but significantly increased by the application of 3 μM α-pompilidotoxin, an effect that was reversed by strophanthidin. The data were compared using a non-parametric repeated measures ANOVA (Friedman test) with a Dunn post-test ($N_{neurons}$ = 66, $N_{slices}$ = 10 and $N_{mice}$ = 5).

The online version of this article includes the following figure supplement(s) for figure 6:

**Figure supplement 1.** $NADH_{CYT}$ transients in $0Ca^{2+}$-ACSF with boosted $Na^+$ influx are briefer than those in control ACSF.

**Figure supplement 2.** The late overshoot in the NAD(P)H signal disappears in a $Ca^{2+}$-free solution and is not recovered by α-pompilidotoxin application.

**Figure supplement 3.** Spontaneous oscillations in the Peredox signal may occur during the prolonged application of zero $Ca^{2+}$-ACSF, in the presence of EGTA and LDH inhibition.

We studied the role of mitochondrial $Ca^{2+}$ elevations by knocking down the mitochondrial calcium uniporter (*Mcu*) gene in DGCs, which nearly abolished the $Ca^{2+}$ entry into neuronal mitochondria, as seen also by *Ashrafi et al., 2020*. This diminished mitochondrial $Ca^{2+}$ entry caused a strong attenuation of the reductive phase of the NAD(P)H autofluorescence signal (i.e. the overshoot after initial dip), consistent with a key role for mitochondrial $Ca^{2+}$ in activating several dehydrogenases in the TCA cycle (reviewed by *McCormack et al., 1990*).

The absence of MCU, however, only marginally decreased the prompt oxidation of mitochondrial NADH upon stimulation (the rapid dip). This suggests that the direct contribution of mitochondrial $Ca^{2+}$ to mitochondrial depolarization and subsequent NADH-fueled proton pumping is negligible compared to that of ADP, which is produced by $Na^+$, $K^+$, and $Ca^{2+}$ pumping at the plasma membrane during periods of activity. Our results, however, contrast with *Duchen, 1992*, who completely abolished both phases of the NAD(P)H signal in isolated neurons by blocking the MCU with ruthenium red. This compound interferes with $Ca^{2+}$ pumping and, thus, ADP production (*Watson et al., 1971*), so it is possible that these effects might have led to the absence of an initial dip in Duchen's study.

Removing $Ca^{2+}$ from the extracellular solution recapitulated the effects seen with *Mcu* knockdown: the dip of the NAD(P)H signal was only decreased by ~21% while the overshoot was almost abolished, indicating again that $Ca^{2+}$ influx is essential for the excess in NADH production in the mitochondria. Using a similar manipulation, *Duchen, 1992* abrogated the full signal, while *Kann et al., 2003* reported only a partial decrease (~59%) in the overshoot, and *Shuttleworth et al., 2003* observed the preservation of the full NAD(P)H transient. It is possible that differences between cultured cells, acute or organotypic brain slices, cell types, or even the stimulation paradigm, may contribute to differences among studies. Apparently, the NAD(P)H overshoot can be $Ca^{2+}$-independent in certain experimental circumstances, possibly produced by rapid pyruvate uptake into the mitochondria upon stimulation, rather than TCA stimulation by $Ca^{2+}$ (*Baeza-Lehnert et al., 2019*).

## OXPHOS is largely preserved in neurons despite the loss of the NAD(P)H overshoot

Although the NAD(P)H overshoot has been universally observed in metabolic studies with neuronal stimulation, it is apparently not essential for maintenance of oxidative phosphorylation. We found that oxygen utilization, a sensitive indicator of flux through OXPHOS (*Hall et al., 2012*; *Ivanov et al., 2014*), was only slightly diminished or delayed by MCU knockdown (depending on the duration of the stimulus), even though the NAD(P)H overshoot is practically eliminated. This agrees with results on isolated brain and heart mitochondria from MCU-KD animals (*Szibor et al., 2020*). What accounts for the continued ability of mitochondria to engage in OXPHOS? Of course, an overshoot of NADH is not required to prevent mitochondrial NADH levels from becoming limiting for OXPHOS; all that is required is maintenance of NADH levels. Even without augmented TCA cycle production of NADH, the increased supply of reducing equivalents produced in glycolysis and

transferred to mitochondria via the malate-aspartate shuttle (MAS) may be substantial. Both the diminished mitochondrial $Ca^{2+}$ due to knockdown of MCU and the still-increased cytosolic $Ca^{2+}$ will contribute to increased MAS function (*Pardo et al., 2006*; *Bak et al., 2012*; *Llorente-Folch et al., 2013*).

Mitochondrial fuels whose metabolism is less dependent on $Ca^{2+}$ stimulation may also contribute to production of mitochondrial NADH. For instance, we find that in contrast to the NAD(P)H over-shoot, the late reductive phase of increased $FADH_2$ (seen as a prolonged decrease in $FAD^+$ auto-fluorescence) is preserved after MCU knockdown or when the MPC is blocked (*Figure 3—figure supplement 3*, and *Figure 2—figure supplements 2* and *3*, respectively). Pyruvate unavailability may be partially compensated by the use of alternative mitochondrial fuels, such as glutamate and glutamine (*Tildon et al., 1985*; *McKenna et al., 1993*; *Westergaard et al., 1995*; *Olstad et al., 2007*; *Divakaruni et al., 2017*). Indeed, these amino acids have been shown to sustain flux through the TCA cycle from $\alpha$-ketoglutarate to oxaloacetate, ultimately resulting in aspartate accumulation as a result of oxaloacetate transamination (*Erecińska et al., 1988*; *Erecińska et al., 1990*; *Sonnewald and McKenna, 2002*; reviewed by *McKenna, 2007*).

The preservation of the $O_2$ transients in the DGC layer contrasts with the previously reported impairment of ATP production in MCU-deficient axon terminals during sustained activity (*Ashrafi et al., 2020*). The contributions of MCU and the NAD(P)H overshoot to OXPHOS, as well as to glutathione regeneration, may differ among cellular compartments (i.e. somata, dendrites/spines, and axons/synaptic terminals), especially when facing high energy demands.

## Neuronal glycolysis is triggered not by $Ca^{2+}$ signaling but rather by ATP hydrolysis

In addition to OXPHOS, neurons also respond to stimulation by increasing the rate of glycolysis, which we found to be strongly associated with the rise in $[Ca^{2+}]_{CYT}$. We considered a potential role for $Ca^{2+}$ acting as a feedforward signal through protein kinases, which may trigger ATP production in anticipation for future episodes of activity and energy demand, but we tested multiple known $Ca^{2+}$ signaling pathways and were unable to find one that was important for the glycolytic response.

We pharmacologically inhibited the $Ca^{2+}$/CaM signaling pathway, which led to increased $[Ca^{2+}]_{CYT}$ at baseline, probably reflecting some inhibition of the 'housekeeping' $Ca^{2+}$-ATPase isoform PMCA1 (*Brini et al., 2013*). However, the NADH/NAD$^+$ ratio remained responsive to rises in the $[Ca^{2+}]_{CYT}$ in the presence of either of the two inhibitors tested (E6-berbamine and calmidazolium). In fact, the amplitude of the $NADH_{CYT}$ transient relative to the $Ca^{2+}$ spike was even higher when the $Ca^{2+}$/CaM complex was inhibited, indicating that this signaling pathway is not responsible for the glycolytic response to acute stimulation.

We also explored a connection between $Ca^{2+}$ and AMPK, a protein kinase that promotes glucose utilization (*Marsin et al., 2000*; *Wu et al., 2013*) and can be activated by CaMKK (*Hawley et al., 2005*; *Woods et al., 2005*). More importantly, its main activator, AMP, reflects the depletion of the ATP pool (reviewed by *Herzig and Shaw, 2018*), which can be a consequence of neuronal activity (*Gerkau et al., 2019*). However, even though AMPK seems poised for the integration of two signals derived from acute neuronal stimulation, its inhibition did not prevent the glycolytic $NADH_{CYT}$ transients in the soma of DGCs. Overall, AMPK seems irrelevant for neuronal lactate production, as observed in cultured cortical neurons (*Muraleedharan et al., 2020*), however, it promotes glucose uptake and glycolysis during prolonged periods of activity in synaptic terminals (*Ashrafi et al., 2017*), suggesting that AMPK signaling may be tailored to cope with local energy demands.

A more universal and ancient mechanism for adjusting energy supply in the face of demand could be the direct activation of glycolysis by consumption of ATP. The buildup of ADP and/or AMP can directly activate phosphofructokinase (PFK) (*Passonneau and Lowry, 1962*; *Erecińska and Silver, 1989*) and drive glycolysis. These mechanisms could act in neurons despite their low levels of Pfkfb3, the enzyme that produces the potent PFK activator fructose-2,6-bishosphate (F2,6BP) (*Herrero-Mendez et al., 2009*). This is not unprecedented: in skeletal muscle fibers, failure to increase F2,6BP after repetitive stimulation can be compensated by AMP, ADP, and other allosteric modulators of PFK (*Wegener and Krause, 2002*).

Our experiments provide evidence that neuronal glycolysis can indeed be driven by ATP hydroly-sis rather than by $Ca^{2+}$ signaling: we were able to elicit $NADH_{CYT}$ transients by increasing $Na^+$ influx, in the absence of extracellular $Ca^{2+}$. Calcium is not strictly necessary for producing these glycolytic

transients; the activity of the ATP-consuming ion pumps is sufficient, specifically the $Na^+$,$K^+$-ATPase (NKA; sodium pump) in the case of the $Na^+$-only transients. Under these conditions, the $Na^+$ pump activity is also necessary, as the $Na^+$-driven glycolytic responses were reversed by the sodium pump inhibitor, strophanthidin.

The effectiveness of the low concentration of strophanthidin used here (10 µM) to block the $NADH_{CYT}$ transients suggests that the NKA isoform α3 could mediate the coupling between neuronal activity and glycolysis. This isoform is more expressed in neurons than in any other cell type in the brain (*Zhang et al., 2014*; *Zeisel et al., 2015*; *Hrvatin et al., 2018*), and confers the high sensitivity of axonal preparations to cardiotonic steroids (*Marks and Seeds, 1978*; *Sweadner, 1979*; *Urayama and Sweadner, 1988*; *Sweadner, 1989*). However, given the apparent low affinity of this NKA isoform for $Na^+$ ($K_{0.5}$=30–70 mM; *Munzer et al., 1994*; *Crambert et al., 2000*; *Hamada et al., 2003*), it only seems poised to counteract $Na^+$ accumulation after strong neuronal activity (*Munzer et al., 1994*; *Azarias et al., 2013*), or with the α-pompilidotoxin-induced augmentation of $Na^+$ influx used here.

Calcium is surprisingly important for promoting glycolysis in control ACSF, even though it seems unlikely that $Ca^{2+}$ accounts for most of the total ion pumping after neuronal excitation. It is possible that $Ca^{2+}$ extrusion is better coupled to glycolysis than $Na^+$ extrusion (*Gover et al., 2007*; *Ivannikov et al., 2010*; *Fernández-Moncada and Barros, 2014*), perhaps due to interactions between $Ca^{2+}$ ATPases (PMCAs) and glycolytic enzymes at the plasma membrane (reviewed by *Dhar-Chowdhury et al., 2007*; *Bruce, 2018*; *James et al., 2020*). In addition, this coupling might require a relatively calmodulin-insensitive isoform like PMCA2 (*Elwess et al., 1997*; reviewed by *Brini et al., 2013*), which would be consistent with the preservation of the $NADH_{CYT}$ transient after the application of calmidazolium, a drug that delays $Ca^{2+}$ clearance through PMCAs as a result of inhibiting calmodulin (*Markram et al., 1995*; *Scheuss et al., 2006*). However, although $Ca^{2+}$ ATPases typically present a higher affinity for $Ca^{2+}$ when compared to the $Na^+$/$Ca^{2+}$ exchanger, they also exhibit a lower turnover rate, so their contribution to $Ca^{2+}$ extrusion would depend on the density and localization of transporters in the membrane (*Blaustein and Lederer, 1999*; *Brini and Carafoli, 2011*), and likely on the strength of stimulation as well.

Future work is required to identify the precise routes of $Ca^{2+}$ extrusion coupled to neuronal glycolysis and to fully understand the biological significance of this process. Glycolysis may provide a fast and localized ATP supply via the phosphoglycerate kinase and pyruvate kinase reactions near the site of high energy demand, as well as additional ATP production from the oxidation of reducing equivalents shuttled into the mitochondria. In addition, the GAPDH-derived $NADH_{CYT}$ transients may also reflect some contribution from the pentose phosphate pathway since some intermediates can be re-introduced into the glycolytic pathway, thus providing the much needed antioxidant capacity for neurons (*Herrero-Mendez et al., 2009*).

In summary, our work provides novel evidence on how neurons cope with moment-to-moment energy demands during bouts of action potentials, highlighting the different roles of $Ca^{2+}$ in coordinating increases in the TCA cycle and glycolysis. Considering the differences in morphology, $Ca^{2+}$ buffering capacity, ion channels, and other cellular components, we anticipate variations in the metabolic responses—and their regulation—in small compartments (e.g. dendrites/spines and axons/synaptic terminals), as well as in different cell-types in the brain (e.g. interneurons and astrocytes).

# Materials and methods

## Key resources table

| Reagent type (species) or resource | Designation | Source or reference | Identifiers | Additional information |
|---|---|---|---|---|
| Chemical compound, drug | NBQX (6-Nitro-7-sulfamoylbenzo [f]quinoxaline-2,3-dione, Disodium Salt) | Toronto Research Chemicals | Cat#N550005; CAS:479347-86-9 | |
| Chemical compound, drug | D-AP5 (D-(-)—2-Amino-5-phosphonopentanoic acid) | Abcam | Cat#ab120003; CAS:79055-68-8 | |

*Continued on next page*

*Continued*

| Reagent type (species) or resource | Designation | Source or reference | Identifiers | Additional information |
|---|---|---|---|---|
| Chemical compound, drug | Isradipine | Abcam | Cat#ab120142; CAS:75695-93-1 | |
| Chemical compound, drug | α-pompilidotoxin | Alomone Labs | Cat#P-170 | |
| Chemical compound, drug | EGTA-AM | Anaspec Inc | Cat#AS-84100; CAS:99590-86-0 | |
| Chemical compound, drug | Calmidazolium | Cayman Chemical | Cat#14442; CAS:57265-65-3 | |
| Chemical compound, drug | Dorsomorphin dihydrochloride | Tocris Bioscience | Cat#3093; CAS:1219168-18-9 | |
| Chemical compound, drug | E6-berbamine | Santa Cruz | Cat#sc-221573; CAS:73885-53-7 | |
| Chemical compound, drug | UK5099 | Santa Cruz | Cat#sc-361394; CAS:56396-35-1 | |
| Chemical compound, drug | UK5099 | Tocris Bioscience | Cat#5185 | |
| Chemical compound, drug | GSK-2837808A | Tocris Bioscience | Cat#5189; CAS:1445879-21-9 | |
| Chemical compound, drug | $MgCl_2$ (1M solution) | Teknova | Cat#M0304 | |
| Chemical compound, drug | Picrotoxin | Sigma-Aldrich | Cat#P1675; CAS:124-87-8 | |
| Chemical compound, drug | Sodium pyruvate | Sigma-Aldrich | Cat#P8574; CAS:113-24-6 | |
| Chemical compound, drug | $CdCl_2$ | Sigma-Aldrich | Cat#202908; CAS:10108-64-2 | |
| Chemical compound, drug | Strophanthidin | Sigma-Aldrich | Cat#S6626; CAS:66-28-4 | |
| Strain, strain background *Mus musculus* | C57BL/6NCrl mice | Charles River | RRID:IMSR_CRL:27 | |
| Strain, strain background *Mus musculus* | *Edil3$^{Tg(Sox2-cre)1Amc}$*/J mice (Sox2-Cre) | The Jackson Laboratory | RRID:IMSR_JAX:004783 | |
| Strain, strain background *Mus musculus* | *Mcu$^{fl/fl}$* mice | *Kwong et al., 2015* | RRID:IMSR_JAX:029817 | |
| Strain, strain background *Mus musculus* | *Dock10Cre* mice | *Kohara et al., 2014* | | Tonegawa lab |
| Strain, strain background *Mus musculus* | *Mcu$^{fl/\Delta}$* mice | This paper | | Obtained by germline deletion of the floxed allele, followed by backcrossing; Yellen lab |
| Other | AAV9.Syn.RCaMP1h.WPRE.SV40 | Penn Vector Core | Discontinued | Viral vector |
| Recombinant DNA reagent | AAV.CAG.Peredox.WPRE.SV40 | *Mongeon et al., 2016* | RRID:Addgene_73807 | |
| Recombinant DNA reagent | AAV.Syn.RCaMP1h.WPRE.SV40 | *Akerboom et al., 2013* | | |
| Recombinant DNA reagent | pZac2.1-CaMKII-mito-GCaMP6s | *Li et al., 2014* | | |

*Continued on next page*

*Continued*

| Reagent type (species) or resource | Designation | Source or reference | Identifiers | Additional information |
|---|---|---|---|---|
| Recombinant DNA reagent | AAV.CAG.Pyronic. WPRE.SV40 | *San Martín et al., 2014* | Derived from RRID:Addgene_51308 | |
| Recombinant DNA reagent | AAV.Syn.mito-RCaMP1h.WPRE.SV40 | This paper | | Produced by addition of a mitochondrial targeting signal; Yellen lab |
| Software, algorithm | MATLAB R2014b | Mathworks | RRID:SCR_001622 | |
| Software, algorithm | Origin 9.1 | OriginLab | RRID:SCR_002815 | |
| Software, algorithm | GraphPad Instat version 3.06 | GraphPad Software | RRID:SCR_000306 | |
| Software, algorithm | Microsoft Excel version 2009 | Microsoft | RRID:SCR_016137 | |

## Reagents

All reagents were purchased from Sigma-Aldrich (St. Louis, MO), unless otherwise specified. The synaptic blocker NBQX was obtained from Toronto Research Chemicals (Toronto, ON). The MPC blocker UK5099 was purchased either from Tocris Bioscience (Bristol, UK) or Santa Cruz (Dallas TX). Another drug from Santa Cruz was E6-berbamine, an inhibitor of the $Ca^{2+}$/calmodulin signaling pathway. Calmidazolium, a drug with similar effects on the $Ca^{2+}$/calmodulin complex, was obtained from Cayman Chemical (Ann Arbor, MI). The L-type calcium channel blocker isradipine and the NMDA-glutamate receptor inhibitor D-AP5 were obtained from Abcam (Cambridge, MA). Stock solutions of $MgCl_2$ (1M) were purchased from Teknova (Hollister, CA). dorsomorphin dihydrochloride (Compound C) and GSK-2837808A were obtained from Tocris (Bristol, UK), EGTA-AM from Anaspec Inc (Fremont, CA) and α-pompilidotoxin from Alomone Labs (Jerusalem, Israel).

We prepared stock solutions of calmidazolium (100 mM), E6-berbamine (33 mM), EGTA-AM (100 mM), GSK-2837808A (10 mM), isradipine (50 mM), strophanthidin (500 mM), and UK5099 (20 mM) in DMSO. The final concentration of DMSO in the experiments was kept ≤0.04%, except for EGTA-AM (0.1%), for which control experiments with 0.1% DMSO-only solution were performed to rule out interferences from the organic solvent in the recordings.

We used $Cd^{2+}$ as a non-selective blocker of high voltage activated $Ca^{2+}$ channels (*Swandulla and Armstrong, 1989*; *Catterall et al., 2005*), at a concentration that avoid significant off-target effects on $Na^+$ channels (*DiFrancesco et al., 1985*) or the $Na^+/Ca^{2+}$ exchanger (*Hobai et al., 1997*).

Experiments with sequential application of α-pompilidotoxin and strophanthidin were challenging because the prolonged exposure to a nominal $Ca^{2+}$-free solution, plus LDH inhibition, caused spontaneous elevations of the Peredox lifetime in some cells (*Figure 6—figure supplement 3*). Furthermore, in slices treated with α-pompilidotoxin, extensive inhibition of the $Na^+$ pumps caused cell swelling and death upon stimulation. It was necessary to lower the concentrations of both α-pompilidotoxin and strophanthidin to ensure the preservation of neuronal viability.

## Animals

Brain slice experiments were performed using brains of male and female wild-type mice (C57BL/6NCrl; Charles River Laboratories), $Mcu^{fl/fl}$, $Mcu^{fl/\Delta}$ and $Mcu^{fl/\Delta}$ *Dock10Cre* (see below). Animals were housed in a barrier facility in individually ventilated cages with ad libitum access to standard chow diet (PicoLab 5053). All experiments were performed in compliance with the NIH Guide for the Care and Use of Laboratory Animals and the Animal Welfare Act. The Harvard Medical Area Standing Committee on Animals approved all procedures involving animals (Animal Welfare Assurance Number A3431-01, Protocol IS00001113-3).

## Generation of $Mcu^{fl/\Delta}$ and $Mcu^{fl/\Delta}$ Dock10Cre mice

We selectively inactivated the *Mcu* gene in DGCs by crossing conditional knockout $Mcu^{fl/fl}$ mice (*Kwong et al., 2015*) with a Cre-driver line specific to DGCs (*Kohara et al., 2014*). Male $Mcu^{fl/fl}$ mice (*Kwong et al., 2015*) were crossed with female Dock10Cre mice (*Kohara et al., 2014*) to generate $Mcu^{fl/+}$ Dock10Cre mice, which were backcrossed with $Mcu^{fl/fl}$ mice to obtain $Mcu^{fl/fl}$ Dock10-Cre mice. Additional adjustments were necessary to maximize the consistency of the MCU-KD phenotype. First, we produced hemizygous $MCU^{fl/\Delta}$ mice so that strong knockdown of *Mcu* would require deletion of only a single copy of the gene to compensate for the limited efficiency of Cre-dependent recombination (*Bao et al., 2013*). Male $Mcu^{fl/fl}$ mice were mated with female $Sox2\text{-}Cre^{+/-}$ mice (*Hayashi et al., 2002*; obtained from The Jackson Laboratory) to generate an offspring with 50% of $Mcu^{wt/\Delta}$ mice (without the Sox2-Cre transgene). Finally, the crossing of male $Mcu^{wt/\Delta}$ mice with female $Mcu^{fl/fl}$ Dock10Cre mice produced a 25% offspring of each experimental genotype: $Mcu^{fl/\Delta}$ and $Mcu^{fl/\Delta}$ Dock10Cre (with *Mcu* selectively deleted in DGCs). Other combinations from the above-mentioned genotypes were able to produce experimental mice (although with different proportions). In all crossings, the Dock10Cre transgene was present in the female parent.

We also worked with adult mice ($82 \pm 12$ days-old, N = 78) instead of juveniles to permit more complete knockdown in the face of the late, postnatal expression of the Dock-10 promoter (*Jaudon et al., 2015*) and the slow turnover of mitochondrial proteins (with a half-time of ~25 days; *Beattie et al., 1967*; *Menzies and Gold, 1971*).

## DNA plasmids and viral vectors

### Construction of AAV.Syn.mito-RCaMP1h.WPRE.SV40

We created an AAV plasmid that targets the sensor RCaMP1h into the mitochondrial matrix by inserting a portion of the precursor of the mitochondrial protein COX8 in the N-terminus of the fluorescent protein, as previously reported for GFP (*Rizzuto et al., 1995*) and GCaMP6s (*Li et al., 2014*).

A 101 bp EcoRI-BamHI DNA fragment from pZac2.1-CaMKII-mito-GCaMP6s (obtained from Dr. Shinghua Ding), containing the mitochondrial leader sequence, was subcloned into the backbone of EcoRI-BamHI DNA fragment (5569 bp) of AAV.Syn.RCaMP1h.WPRE.SV40 (obtained from Dr. Loren Looger), using T4 DNA ligase.

### Construction of AAV.CAG.Pyronic.WPRE.SV40

We cloned the pyruvate sensor Pyronic into an AAV plasmid with the CAG promoter. The Pyronic insert, 2272 bp BamHI-HindIII DNA fragment, from Pyronic/pcDNA3.1(-) (Addgene# 51308, deposited by *San Martín et al., 2014*) was subcloned into the backbone of BamHI-HindIII DNA fragment (4696 bp) of AAV.CAG.Laconic.WPRE.SV40 (AAV construct from *Díaz-García et al., 2017*, derived from *San Martín et al., 2013* - Addgene# 44238), using T4 DNA ligase. The resultant plasmid, AAV.CAG.Pyronic.WPRE.SV40, was used to clone in an optimal Kozak sequence (CCACC). A 771 bp amplicon containing the optimized Kozak sequence was generated by using PCR and was amplified from pcDNA3.1-Pyronic (Forward Primer: 5' GAATTGGATCCACCATGGTGAGCAAGGGCGAGGA-GAC 3' and Reverse Primer: 5' GGGCGAATTCGGAGCCTGC 3'). The amplicon was double digested with BamHI-EcoRI and was subcloned, using T4 DNA ligase, into the backbone of BamHI-EcoRI DNA fragment (6211 bp) of AAV.CAG.Pyronic.WPRE.SV40.

### Production of AAV particles

Custom-made adeno-associated vectors (AAV) were used for biosensor expression in brain tissue. For the expression of Peredox in the hippocampus, we used the AAV8 serotype (obtained from the Penn Vector Core, University of Pennsylvania, PA) and the universal promoter CAG (*Mongeon et al., 2016*). For expression of the pyruvate sensor Pyronic, we used the AAV9 serotype (obtained from the Viral Core Facility from Children Hospital in Boston, MA) and the universal promoter CAG.

For expression of the $Ca^{2+}$ sensor RCaMP1h, we used the AAV9 serotype and the neuron-specific promoter synapsin. For 2p-FLIM experiments, we used viral batches from three different suppliers: the Penn Vector Core, University of Pennsylvania, PA, the Viral Core Facility from Children Hospital in Boston, MA, and the Center for Genomics and System Biology, New York University, Abu Dhabi, UAE (kindly provided by Dr. G. Fishell and Dr. J Dimidschstein). For $Ca^{2+}$ imaging in

autofluorescence experiments, AAV9.Syn.RCaMP1h.WPRE.SV40 was produced in our laboratory using a protocol reported elsewhere (*Kimura et al., 2019*).

## Biosensor expression in hippocampus

For sensor expression in the hippocampus, mice at postnatal day 1 or 2 were anesthetized using cryoanesthesia. Following confirmation of anesthesia, the viral mix was loaded onto a pulled glass capillary pipette (Wiretrol II, Drummond Scientific Company, Broomall, PA) and the pups were intracranially injected with 150 nl of the AAV mix, twice per hemisphere, at the following coordinates with respect to lambda: (i) 0 mm in the anterior-posterior direction, ±1.9 mm in the medial-lateral axis, and −2.0 mm in the dorsal-ventral direction (ii) 0 mm in the anterior-posterior direction, ±2.0 mm in the medial-lateral axis and −2.3 mm in the dorsal-ventral direction. Viral injections were delivered at a rate of 50 nl/min using an UltraMicroPump III (WPI, Sarasota, FL) microinjector. After injections, we waited ~2 min before gently pulling out the pipette, as a precaution to avoid spilling virus outside the target area. The pups recovered on a heat pad (covered by a paper towel) and/or under a heat lamp, before returning them to their cages. As a post-surgery care, one subcutaneous injection of ketoprofen (10 mg/kg) was delivered for up to 3 days. Acute brain slices were suitable for imaging from 2 weeks to 4 months post-injection.

Alternatively, some intracranial injections were performed in adult mice (after postnatal day 45). Mice were administered dexamethasone sodium phosphate (8 mg/kg) by an intramuscular injection to the hind leg, 1–2 hr before the surgery. Animals were anesthetized with isoflurane (induction: 4–5%, maintenance: 1–3%). Following confirmation of anesthesia, the mice were placed on a heated pad to maintain the body temperature at 37°C. Local anesthetics (10 mg/kg lidocaine and 2.5 mg/kg bupivacaine) were injected subcutaneously at the incision site, and the analgesic ketoprofen (10 mg/kg) and 0.5 ml of sterile 0.9% NaCl solution were also injected subcutaneously prior to the surgery. A small incision was performed in the skin to expose the skull, and a small hole was drilled over the desired area on the right hemisphere. The stereotactic coordinates with respect to lambda, for a single 2 µl viral injection, were the following: 3.39 mm in the anterior-posterior direction, −2.2 mm in the medial-lateral axis, and −2.4 mm in the dorsal-ventral direction. At the end of the surgery, a subcutaneous injection of buprenorphine SR (0.75 mg/kg) was delivered. We waited at least 2 weeks post-injection for experiments in acute hippocampal slices.

For 2p-FLIM experiments, the viral titer (genome copies/ml) used for injecting AAV8.CAG.Pyronic.WPRE.SV40 was $1.64 \times 10^{14}$ gc/ml, and $1.45 \times 10^{13}$ gc/ml for AAV9.Syn.mito-RCaMP1h.WPRE.SV40 (BCH). The viral titers in the Peredox:RCaMP1h mix varied depending on the AAV providers (in parenthesis): $2.05 \times 10^{12}$ gc/ml of AAV8.CAG.Peredox.WPRE.SV40 (UPenn) and $1.87 \times 10^{10}$ gc/ml of AAV9.Syn.RCaMP1h.WPRE.SV40 (Abu Dhabi), $4.1 \times 10^{12}$ gc/ml of AAV8.CAG.Peredox.WPRE.SV40 (UPenn) and $2.55 \times 10^{14}$ gc/ml of AAV9.Syn.RCaMP1h.WPRE.SV40 (BCH), or $3.08 \times 10^{12}$ gc/ml of AAV8.CAG.Peredox.WPRE.SV40 (UPenn) and $2.01 \times 10^{13}$ gc/ml of AAV9.Syn.RCaMP1h.WPRE.SV40 (UPenn).

For autofluorescence experiments, the viral stocks were diluted in sterile 0.9% NaCl solution prior to the injections, to achieve the following titers: $1.45 \times 10^{13}$ gc/ml for AAV9.Syn.RCaMP1h.WPRE.SV40 (laboratory-made), and $4.83 \times 10^{12}$ gc/ml for AAV9.Syn.mito-RCaMP1h.WPRE.SV40 (BCH). Several dilutions were tested until the bleedthrough of RCaMP fluorescence into the green channel was negligible.

## Mouse hippocampal slice preparation

Mice between 14 and 24 days-old were anesthetized with isoflurane, decapitated, and the brain was placed in ice-cold slicing solution containing (in mM): 87 NaCl, 2.5 KCl, 1.25 NaH$_2$PO$_4$, 25 NaHCO$_3$, 0.5 CaCl$_2$, 7 MgCl$_2$, 75 sucrose, and 25 D-glucose (335–340 mOsm/kg). A different slicing solution was used for 2–4 months-old adult mice (*Ting et al., 2014*), consisting of (mM): 93 N-Methyl-D-Glucamine, 2.5 KCl, 1.2 NaH$_2$PO$_4$, 30 NaHCO$_3$, 20 HEPES, 10 MgSO$_4$, 0.5 CaCl$_2$, 25 D-glucose, 5 sodium ascorbate, 2 thiourea, 3 sodium pyruvate (~310 mOsm/kg; pH 7.4 adjusted with HCl).

Brains were glued by the dorsal side on a specimen tube holder and embedded in warm PBS with 2% low-melting agarose. The agarose was quickly congealed using a chilling block. The specimen tube was inserted in a chamber containing the same slicing solution (previously oxygenated) and horizontal slices were cut at a thickness of 275 µm using a Compresstome slicer (VF-300-0Z,

Precisionary, Natick, MA). Alternatively, brains were glued by the dorsal side in a chamber containing the same slicing solution and horizontal slices were cut at a thickness of 275 µm using a vibrating slicer (7000smz-2, Campden Instruments, Loughborough, England).

Slices were immediately transferred to a chamber filled with artificial cerebrospinal fluid (ACSF) at 37°C, containing (in mM): 120 NaCl, 2.5 KCl, 1 NaH$_2$PO$_4$, 26 NaHCO$_3$, 2 CaCl$_2$, 1 MgCl$_2$, and 10 D-glucose (~290 mOsm/kg). All solutions were continuously bubbled with a mix of 95% O$_2$ and 5% CO$_2$, for adequate oxygenation and pH buffering around 7.4. Slices were incubated at 37°C for 35 min and then at room temperature for at least 30 min before the experiments, which were executed in the next 5 hr after slicing.

The Ca$^{2+}$-free ACSF contained (in mM): 120 NaCl, 2.5 KCl, 1 NaH$_2$PO$_4$, 26 NaHCO$_3$, 1 EGTA, 4.38 MgCl$_2$ (~4.1 free) and 10 D-glucose. In these experiments, the control solution was modified to include EGTA, as follows (in mM): 120 NaCl, 2.5 KCl, 1 NaH$_2$PO$_4$, 26 NaHCO$_3$, 1 EGTA, 3 CaCl$_2$ (~2 free as in regular ACSF), 1 MgCl$_2$, and 10 D-glucose. The concentrations for divalent ions in the presence of EGTA were estimated using Chelator (*Schoenmakers et al., 1992*), through the Maxchelator website (https://somapp.ucdmc.ucdavis.edu/pharmacology/bers/maxchelator/CaMgATPEGTA-TS.htm). An alternative, simpler, nominal zero Ca$^{2+}$ ACSF was also tested (in mM): 120 NaCl, 2.5 KCl, 1 NaH$_2$PO$_4$, 26 NaHCO$_3$, 4.1 MgCl$_2$, and 10 D-glucose. The regular 2 mM CaCl$_2$ was substituted by 3.1 free MgCl$_2$ to preserve the charge screening effect of divalent ions on the membrane surface (*Hille et al., 1975*).

For experiments, a brain slice was attached to a poly-lysine coated coverslip and the recordings were performed in a chamber with a continuous supply of oxygenated ACSF at a flow rate of 5 ml/min. The solution was maintained at 33–34°C using inline heaters (Warner Instruments, Hamden, CT) or custom-made heaters. To prevent degassing in perfusion line, solutions were preheated at 38°C in a waterless bead bath (Cole-Parmer, Vernon Hills, IL) for 2p-FLIM experiments, or in water bath (VWR) for autofluorescence experiments.

DGCs were stimulated with a concentric bipolar electrode CBBEC75 (FHC, Bowdoin, ME) placed in the hilus. During the experiments, the ACSF contained NBQX (5 µM), D-AP5 (25 µM), and picrotoxin (100 µM) to block synaptic activity. Stimulation was delivered in trains of 25–200 brief (0.1 ms) pulses at a frequency of 50 Hz, using an A360 stimulus isolation unit (WPI, Sarasota, FL). The stimulation intensity was adjusted to reliably evoke spike activity, which typically ranged from 750—1500 µA for antidromic stimulation (in autofluorescence experiments, the stimulation intensity was always set at 1000 µA).

## Two-photon fluorescence lifetime imaging microscopy

Lifetime imaging data were acquired with a modified Thorlabs Bergamo II microscope (Thorlabs Imaging Systems, Sterling, VA), with hybrid photodetectors R11322U-40 (Hamamatsu Photonics, Shizuoka, Japan); the light source was a Chameleon Vision-S tunable Ti-Sapphire mode-locked laser (80 MHz,~75 fs; Coherent, Santa Clara, CA). The objective lens used for brain slice imaging was an Olympus LUMPLFLN 60x/W (NA 1.0). An excitation wavelength of 790 nm was used for the Peredox and RCaMP sensors. Fluorescence emission light was split with an FF562-Di03 dichroic mirror and bandpass filtered for green (FF01-525/50) and red (FF01-641/75) channels (all filter optics from Semrock, Rochester, NY). For the Pyronic sensor, excitation was at 850 nm, and emission light was split with an FF506-Di03 dichroic mirror and bandpass filtered for CFP (FF01-475/35) and YFP (FF01-542/27) channels. The photodetector signals and laser sync signals were preamplified and then digitized at 1.25 gigasamples per second using a field programmable gate array board (PC720 with FMC125 and FMC122 modules, 4DSP, Austin, TX).

Laboratory-built firmware and software performed time-correlated single photon counting to determine the arrival time of each photon relative to the laser pulse; the distribution of these arrival times indicates the fluorescence lifetime (*Yellen and Mongeon, 2015*; *Mongeon et al., 2016*). Lifetime histograms were fitted using nonlinear least-squares fitting in MATLAB (Mathworks, Natick, MA), with a two-exponential decay convolved with a Gaussian for the impulse response function (*Yasuda et al., 2006*). Microscope control and image acquisition were performed by a modified version of the ScanImage software written in MATLAB (*Pologruto et al., 2003*; provided by B. Sabatini and modified by G.Y.).

## Lifetime imaging quantification

Image analysis was performed using MATLAB software developed in our laboratory. Regions of interest (ROIs) were defined around individual cells, and photon statistics were calculated for all pixels within the ROI. Typical ROIs encompassed 100–900 pixels, in images of 128 × 128 pixels acquired at a scanning rate of 2 ms per line. Lifetime values were calculated as a standardized 'tau8' value, which minimizes the variability of the fits by restricting the averaging to the approximate time window of the actual data (*Díaz-García et al., 2019*). Most data points in the time series plots of lifetimes are for the mean value of 20 sequentially acquired frames, except that for RCaMP1h (Calcium) data in the 10 s after stimulation, the data points represent individual frames.

Bleedthrough of green Peredox fluorescence into the red RCaMP optical channel was corrected using the ratio of measured red to green fluorescence observed when only Peredox was expressed, approximately 6.0% (*Díaz-García et al., 2017*). Similarly, RCaMP1h expression alone led to some signal in the green optical channel, probably due to an immature fluorophore, in direct experiments approximately 4.4% of the red fluorescence intensity, and having a tau8 of ~0.675 ns; these values were used to correct the baseline in dual expression experiments.

Neurons with sudden and irreversible increases in the baseline lifetime of RCaMP or Peredox were excluded from the analysis.

## Widefield fluorescence microscopy and electrical stimulation

Autofluorescence signals in brain slices (also expressing a red $Ca^{2+}$ biosensor, RCaMP1h or mito-RCaMP1h) were visualized with an Olympus BX51WI upright microscope using an LUMPlanFl/IR 60x/0.90W objective. The excitation light was delivered by an AURA light engine (Lumencor, Beaverton, OR) at 365, 480 and 575 nm to excite NAD(P)H, $FAD^+$ and the red $Ca^{2+}$ biosensor, respectively. The times of exposure were 100 ms (excitation at 365 nm), 50—85 ms (excitation at 480 nm), and 5—85 ms (excitation at 575 ms). The fluorescence emission light was split with an FF395/495/610-Di01 dichroic mirror and bandpass filtered with an FF01-425/527/685 filter (all filter optics from Semrock, Rochester, NY). Images were collected with a CCD camera (IMAGO-QE, Thermo Fisher Scientific), at a rate of 1 frame every two seconds in non-stimulated conditions, alternating the excitation wavelengths in each frame. During the fast response to stimulation, the acquisition rate was temporarily increased to six frames per second, for a total duration of 10 s (starting 2 s prior to each stimulation), and then returned to the initial lower frequency acquisition rate. Image acquisition and analysis was performed using laboratory-built software written in MATLAB (Mathworks, Natick, MA). This software communicated with an electrophysiology setup by sending an external trigger to a protocol written in pClamp 10 (Molecular Devices, San Jose, CA), which in turn controlled a A360 stimulus isolation unit (WPI, Sarasota, FL) via a Digidata 1321A digitizer (Molecular Devices, San Jose, CA). In pilot experiments, some recordings of local field potentials in the DGC layer were collected and amplified via a Multiclamp 700B (Molecular Devices, San Jose, CA) to confirm the stimulation, but they were not performed routinely in subsequent experiments.

For analysis, the DGC layer was identified in an image obtained with transmitted light, and an ROI was drawn around the somata. Typical ROIs encompassed 4000–4500 pixels, in images of 172 × 130 pixels acquired at 16 bits per pixel and a binning of 8. The average fluorescence (as Arbitrary Units; A.U.) per pixel was calculated in each ROI (A.U./px), as well as the relative fluorescence intensity changes (ΔF/F).

Unlike many stimulation paradigms used for recording autofluorescence signals in brain slices (*Shuttleworth et al., 2003*; *Brennan et al., 2006*; *Brennan et al., 2007*; *Ivanov et al., 2014*), ours prevented the signaling through ionotropic receptors by direct electrical stimulation of the axons and the inclusion of synaptic blockers in the ACSF. Therefore, we measured signals in response to backpropagating action potentials in the somata. By limiting our analysis to the DGC layer, we minimized the contribution of astrocytes to these signals: this region in the dentate gyrus is densely packed with neurons, and its neuron-to-astrocytes ratio is higher than the average value for the whole hippocampus (*Lana et al., 2017*; *Keller et al., 2018*).

## Extracellular O₂ recordings

These measurements were executed simultaneously with the imaging of the autofluorescence signals. The Clark type oxygen glass microsensors, with tip diameters of 10 or 25 μm (Unisense, Aarhus,

Denmark), were pre-polarized overnight for the initial calibration, which was performed in ACSF saturated with 20% or 95% $O_2$, and an anoxic solution of 0.1 M sodium ascorbate and 0.1M NaOH (following the manufacturer instructions). Subsequent calibrations were performed at least once a week using two points: zero (anoxic solution) and 20% $O_2$ in ACSF, both at 33–34°C. For routine calibrations, the pre-polarization times ranged between 1 and 2 hr, or until the signal was stable for 10 min. The microelectrode was connected to an OXY-Meter amplifier (Unisense, Aarhus, Denmark), and the signal acquired with the Sensor Trace software (Unisense, Aarhus, Denmark) at a rate of 1 Hz. The microsensor tip was inserted ~140 µm into the slice, in the DGC layer, and the signal was calibrated and expressed as $[O_2]$ in µM. The data was exported to an Excel file, and then analyzed offline using laboratory-built software written in MATLAB (Mathworks, Natick, MA). Interpretation of the $O_2$ changes in terms of ATP synthesis are made with the assumption that the proton leak across the mitochondrial membrane (independent of ATP synthesis and $Ca^{2+}$ pumping) is unchanged.

## Statistical analysis

Statistical analyses were performed using GraphPad InStat v3.06 (GraphPad Software, San Diego, CA). Data were tested for normality with a Kolmogorov-Smirnov test. If the data did not fulfill all the assumptions for parametric tests (paired or unpaired Student's t-test, one-way or repeated measures ANOVA with a Student-Newman-Keuls post-test), an equivalent non-parametric test was used. For comparisons of two populations with normal distributions but different SDs, a Welch's t-test was used. The alternatives used in this study were: the Mann-Whitney test (for unpaired comparisons between two groups), the Wilcoxon matched pairs test (similar but for paired comparisons), the Kruskal-Wallis test (for multiple comparisons) and the Friedman test for repeated measures, with a Dunn's post test. The selected tests and post-tests, as well as the descriptive statistics, are indicated in the figure legends and throughout the manuscript. Values are expressed as mean ± SD for individual neurons, except for autofluorescence experiments and $O_2$ recordings, where the standard deviation is calculated using the number of slices.

Graphics were constructed using Origin 9.1 (OriginLab, Northampton, MA) and Microsoft Excel (Microsoft, Redmond, WA). Datasets were represented as box plots comprising the 25—75% (Q2—Q3) interquartile range, with whiskers expanding to the lower (Q1) and upper (Q4) quartiles of the distribution (5—95%). The median of the distribution is represented by a horizontal line inside the box, and the mean is represented by a cross symbol (×).

## Acknowledgements

We thank the members of the Yellen laboratory for valuable discussions, and to Hannah Zucker for technical assistance. We also thank Dr. Susumu Tonegawa for providing the *Dock10-Cre* transgenic mice; Dr. Jeffery D Molkentin for providing the $Mcu^{fl/fl}$ transgenic mice; the Viral Core of Boston Children's Hospital, the U Penn Vector Core and the Center for Genomics and System Biology, NYU, Abu Dhabi, UAE (as well as Dr. Gordon Fishell and Dr. Jordane Dimidschstein) for packaging of AAVs; the GENIE project of HHMI Janelia Research Campus, and Dr. Loren Looger and Dr. Douglas Kim for the RCaMP1h sensor; Dr. L Felipe Barros for the Pyronic sensor; and Dr. Shinghua Ding for the mito-GCaMP6s plasmid.

## Additional information

### Competing interests

Gary Yellen: Reviewing editor, *eLife*. The other authors declare that no competing interests exist.

### Funding

| Funder | Grant reference number | Author |
| --- | --- | --- |
| National Institute of Neurological Disorders and Stroke | R01 NS102586 | Gary Yellen |
| National Institute of General Medical Sciences | R01 GM124038 | Gary Yellen |

| NIH Office of the Director | DP1 EB016986 | Gary Yellen |
| National Institute of Neurological Disorders and Stroke | F32 NS100331 | Carlos Manlio Díaz-García |
| National Institute of Neurological Disorders and Stroke | F32116105 | Dylan J Meyer |
| Department of Neurobiology, Harvard Medical School | Fix Fund Postdoctoral Fellowship | Carlos Manlio Díaz-García |
| Department of Neurobiology, Harvard Medical School | Mahoney Postdoctoral Fellowship | Dylan J Meyer |

The funders had no role in study design, data collection and interpretation, or the decision to submit the work for publication.

### Author contributions
Carlos Manlio Díaz-García, Conceptualization, Software, Formal analysis, Funding acquisition, Investigation, Methodology, Writing - original draft, Writing - review and editing; Dylan J Meyer, Conceptualization, Investigation, Writing - review and editing; Nidhi Nathwani, Resources, Methodology, Writing - review and editing; Mahia Rahman, Resources, Investigation, Writing - review and editing; Juan Ramón Martínez-François, Software, Writing - review and editing; Gary Yellen, Conceptualization, Software, Formal analysis, Supervision, Funding acquisition, Methodology, Writing - review and editing

### Author ORCIDs
Carlos Manlio Díaz-García (iD) https://orcid.org/0000-0002-4352-2496
Dylan J Meyer (iD) http://orcid.org/0000-0001-8453-3813
Mahia Rahman (iD) http://orcid.org/0000-0001-6870-4221
Juan Ramón Martínez-François (iD) https://orcid.org/0000-0002-1035-2574
Gary Yellen (iD) https://orcid.org/0000-0003-4228-7866

### Ethics
Animal experimentation: All experiments were performed in compliance with the NIH Guide for the Care and Use of Laboratory Animals and the Animal Welfare Act. The Harvard Medical Area Standing Committee on Animals approved all procedures involving animals. (Animal Welfare Assurance Number A3431-01, Protocol IS00001113-3).

### Decision letter and Author response
Decision letter https://doi.org/10.7554/eLife.64821.sa1
Author response https://doi.org/10.7554/eLife.64821.sa2

## Additional files

### Supplementary files
• Transparent reporting form

### Data availability
All data generated or analysed during this study are included in the manuscript and supporting files.

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
