## [Decision Letter]

**Acceptance summary:**

In this manuscript, Carlos Manlio Diaz-Garcia et al. investigate potential coordination between responses of energy-regenerating pathways in the cytoplasm (glycolysis) and mitochondria (TCA/OXPHOS) during neuronal activation, focusing on the nature of the trigger for glycolytic activation, which is unknown, and the role of Ca^2+^ in mitochondrial responses, which is a matter of debate. Using acute hippocampal slices as an experimental model, an appropriate ensemble of genetic and pharmacological tools, and NADH- and Ca^2+^-indicators as readouts of metabolic activity and activation status, respectively, these authors found that NADH signals in the cytosol occur independently of those in mitochondria. Using mitochondrial Ca^2+^ uniporter (MCU)-knockout mice, they further demonstrated a direct role for mitochondrial Ca^2+^ in driving TCA metabolism.

**Decision letter after peer review:**

Thank you for submitting your article "The distinct roles of calcium in rapid control of neuronal glycolysis and the tricarboxylic acid cycle" for consideration by *eLife*. Your article has been reviewed by three peer reviewers, one of whom is a member of our Board of Reviewing Editors, and the evaluation has been overseen by Richard Aldrich as the Senior Editor. The following individual involved in review of your submission has agreed to reveal their identity: James Hurley (University of Washington) (Reviewer #2).

The reviewers have discussed the reviews with one another and the Reviewing Editor has drafted this decision to help you prepare a revised submission.

Summary:

In this manuscript, Carlos Manlio Diaz-Garcia et al. investigate potential coordination between responses of energy-regenerating pathways in the cytoplasm (glycolysis) and mitochondria (TCA/OXPHOS) during neuronal activation, focusing on the nature of the trigger for glycolytic activation, which is unknown, and the role of Ca^2+^ in mitochondrial responses, which is a matter of debate. Using acute hippocampal slices as an experimental model, an appropriate ensemble of genetic and pharmacological tools, and NADH- and Ca^2+^-indicators as readouts of metabolic activity and activation status, respectively, these authors found that NADH signals in the cytosol occur independently of those in mitochondria. Using mitochondrial Ca^2+^ uniporter (MCU)-knockout mice, they further demonstrated a direct role for mitochondrial Ca^2+^ in driving TCA metabolism. However, they found no evidence for a role for Ca^2+^-dependent signaling in driving glycolysis in the cytoplasm. Instead, they propose that the trigger for glycolytic activation in response to neuronal activity is enhanced energy demand resulting from the ATP-dependent operation of surface membrane Ca^2+^ (and Na^+^) extrusion mechanisms. On the basis of these basic relationships, they conclude that Ca^2+^ plays distinct roles in the rapid control of neuronal glycolysis and the mitochondrial TCA cycle.

The results convincingly demonstrate that energetic responses to neuronal activation in the cytoplasm and mitochondria are independent of each other and that mitochondrial Ca^2+^ is important in TCA cycle activation. They are also consistent with the conclusion that the energy required to pump Ca^2+^ and Na^+^ out of the cell ultimately drives glycolysis.

Essential revisions:

1) Figure 1B legend. Please provide a more complete description of this experiment. The bottom trace shows the Ca^2+^ signal so that should be stated in the legend and the exact time of the train of pulses is not shown so it's timing relative to the RCAMP signal should be indicated. Also, there should be a very brief description of the meaning of fluorescence lifetime. I was familiar with this measurement from previous Yellen reports but some readers may not be.

2) Subsection “Calcium entry into mitochondria is required for strong activation of TCA metabolism”. Based on the results in Figure 2A the loss of MCU seems to be having a specific effect. The magnitude of the initial Ca^2+^ increase in response to stimulation seems to be about the same with and without MCU, but without MCU the influx is not sustained. (I think they discuss later that this may be a non-MCU mechanism of Ca^2+^ influx but some comment should be made about this here). Also, the initial baseline (AU per px) is higher in the MCU knockdown neurons – is that just variation or does it reflect a higher resting Ca^2+^ concentration in the matrix?

3) In Figure 2A Ca^2+^ is measured as AU/px not in lifetime, but in panel C the cytosolic Ca is reported in lifetime units. Please comment on why this was done and how that affects the significance and quality of the data and interpretation of the different types of experiments.

4) In Figure 2B – 50% decay times are reported. Is that 50% of the initial spike of Ca? Or is it from a curve fitting to what appears to be a normal exponential decay curve that occurs following the spike. In other words is the decay of mitochondrial Ca biphasic and if so shouldn't both two time constants and their relative contributions be reported?

5) In the MCU knockdown mitochondrial NADP(H) signals in Figure 2A the oxidative transients seem smaller than in the controls. That does not seem representative based on Figure 2—figure supplement 3. To avoid confusion would it be possible to use a more representative experiment for Figure 2A?

6) Subsection “Calcium entry into mitochondria is required for strong activation of TCA metabolism”. I think there should also be some discussion of AGC1 activity being stimulated directly by cytosolic Ca^2+^. This is mentioned briefly later in the Discussion but it seems like it could be a significant factor in these studies and in their interpretation.

7) Subsection “Energy demand from Na^+^ or Ca^2+^ pumping triggers aerobic glycolysis”. Adding strophanthidin in the zero Ca^2+^ shows that ATP consumption stimulates GAPDH activity. However, wouldn't it be also worthwhile to determine if strophanthidin also blocks the cytosolic NADH increase triggered by stimulation when Ca^2+^ is present? In other words can the authors address whether Ca^2+^ does or does not contribute to accelerated glycolysis (independently of increased energy demand)?

8) Discussion. Please address here whether or not Ca^2+^ regulation of AGC1 (i.e. enhanced MAS activity) also could contribute to elevated NADH in the matrix. (this is a reiteration of my comment #6).

9) Discussion: Is the "rapid dip" blocked by strophanthidin?

10) The data presented show that Ca^2+^ is little more than a bystander in the regulation of glycolysis. The "distinct roles" formulation of the paper needs to be walked back.

11) A scheme showing the biochemical linkages and cell-biological relationships (subcellular compartments, etc) under consideration is necessary. Readers should not be made to forage through biochemistry texts and online resources to make sense of the basic questions a paper is attempting to address.

12) It is important to know the extent of MCU protein depletion; but it is acknowledged that this may be technically difficult. Have all the options been exhausted? Because the extent of MCU depletion is not known, references to possible non-uniporter routes of mitochondrial Ca^2+^ entry (subsection “Calcium entry into mitochondria is required for strong activation of TCA metabolism”) are not only speculative but are potentially provocative.

13) The NADHcyto transient: There was a decrease in the NADHcyt transient with MCU depletion that could not be explained by (greater) flux through LDH because it was not "rescued" by the LDH inhibitor. It was stated in the text that the decrease could reflect increased MAS activity; this would constitute a compensation for loss of MCU.

14) NAD(P)H dip: There was a small decrease in the amplitude of the NAD(P)H autofluorescence dip in MCU kd slices, and the kinetics of the dip were altered; these changes, while small, were significant (Figure 2—figure supplement 3A).

15) Likewise, there were small but significant changes in the concomitant O_2_ signal, in both the amplitude and kinetics (specifically, in the "time to O_2_ dip" (Figure 2—figure supplement 3B).

FAD+ undershoot during the NAD(P)H overshoot: In MCU kd slices, there was less of a decrease in the FAD+ undershoot compared to decrease in the NAD(P)H overshoot. What does the FAD+ undershoot reflect: PDH activity (the E3 subunit: FADH2 -> FAD+), SDH activity (FAD+ -> FADH2) or the glycerol-3-P-dehydrogenase (FADH2 -> FAD+)? Though the latter describes the NAD(P)H overshoot phase, the fact that there is a different effect on the NAD(P)H overshoot and FAD+ undershoot suggests that the FAD+ signal does not simply reflect TCA FAD+ -> FADH2 (at SDH), and that what contributes to the FAD+ signal may differ between Ctrl and MCU kd. The latter point would support\ that metabolic fluxes in response to stimuli can be different in MCU kd slices.

16) FAD+ undershoot in MCU kd slices likely does not reflect flux through PDH or through SDH (since otherwise less of an attenuation of the NAD(P)H signal would be expected). So what does the FAD+ undershoot reflect in MCU kd? Increased glycerol-3-P-dehydrogenase activity? If yes, then there can be substantial ROS associated with that activity (see Quinlan 2013 PMID: 24024165), which could provoke additional changes.

17) Discussion: focus is on emphasizing the effect of MCU kd on the NAD(P)H overshoot and the changes to the dip (NAD(P)H and O_2_) and the FAD+ undershoot seems to be deemphasized. At least in the text of Results, a fuller description of the changes in the dip and undershoot would be needed.

18) This dataset might be the first actual evidence that metabolism compensates/responds/is not neutral to MCU loss (beyond the surprising observations in other studies that MCU loss appears to not impact metabolism). Acknowledging this evidence that metabolism is not neutral to MCU loss seems important in light of observations from several studies that MCU loss appears to not alter metabolism, which can lead to the conclusion that MCU plays no role in metabolism.

19) MAS depends on AGC1/2 (aralar/citrin or SLC25A12/SLC25A13) which contain EF-hands that face the IMS and are therefore theoretically exposed to cytosolic Ca^2+^; it would be worth pointing this out, and that the higher cyto Ca^2+^ during the pulses may lead to greater activation of MAS. The same would be the case for glycerol-3-P-dehydrogenase if it were to play a role.

20) Referring to the NAD(P)H overshoot as TCA-refilling may be incorrect, and prematurely elevates an observation into an interpretation. That there is an NAD(P)H overshoot suggests that stimulation is followed by a phase in which the mitochondrial matrix is in a relatively reduced state. This may reflect a disproportionate NADH generation by PDH and TCA cycle relative to NADH oxidation by the electron transport chain. The overshoot could reflect a shift in oxidation state of the NADPH pool.

21) About the mixed results in the literature on the effects of mitochondrial Ca^2+^ the different phases of NAD(P)H response to stimulation: That there are "contradictory results" is "merely" mentioned in Results and in Discussion. Has the current study been designed in any way that attempts to better understand the role of mito Ca^2+^ in the NAD(P)H response?

22) Discussion: "We found that oxygen utilization,.…, was not altered by MCU knockdown". This does not seem to be accurate based on the data of Figure 2—figure supplement 3.

The sentence "More importantly, Ca^2+^ itself may not be necessary if.… Ca^2+^ extrusion". Metabolic response to Ca^2+^ extrusion and the use of/requirement for Ca^2+^ as part of that metabolic response can be independent.

---

## [Author Response]

Essential revisions:1) Figure 1B legend. Please provide a more complete description of this experiment. The bottom trace shows the Ca^2+^ signal so that should be stated in the legend and the exact time of the train of pulses is not shown so it's timing relative to the RCAMP signal should be indicated.

We have modified the legend to have a more complete description of the experiment, including the RCaMP Ca^2+^ signals, and we have now shown the stimulation times with arrows in the figure.

Also, there should be a very brief description of the meaning of fluorescence lifetime. I was familiar with this measurement from previous Yellen reports but some readers may not be.

Thank you for this suggestion. A brief description of fluorescence lifetime has been included in the Results.

2) Subsection “Calcium entry into mitochondria is required for strong activation of TCA metabolism”. Based on the results in Figure 2A the loss of MCU seems to be having a specific effect. The magnitude of the initial Ca^2+^ increase in response to stimulation seems to be about the same with and without MCU, but without MCU the influx is not sustained. (I think they discuss later that this may be a non-MCU mechanism of Ca^2+^ influx but some comment should be made about this here). Also, the initial baseline (AU per px) is higher in the MCU knockdown neurons – is that just variation or does it reflect a higher resting Ca^2+^ concentration in the matrix?

We have changed the analysis and presentation of the mitochondrial Ca^2+^ signals to better accord with the reviewers’ correct understanding of the effect of MCU-KD. Instead of “decay time”, we now report area-under-the-curve for the mitochondrial Ca^2+^ signals, which gives an integrated measure of both the amplitude and the sustained time course.

We have also performed additional experiments using two-photon fluorescence lifetime measurements of the RCaMP signal in MCU-KD and control neurons (Figure 3—figure supplement 2). These give results that are completely consistent with the widefield fluorescence data, but they also give a more reliable indication of baseline mitochondrial [Ca^2+^] and show that this is in fact lower in the MCU-KD neurons, as might be expected. We now describe this result in the Results.

3) In Figure 2A Ca^2+^ is measured as AU/px not in lifetime, but in panel C the cytosolic Ca is reported in lifetime units. Please comment on why this was done and how that affects the significance and quality of the data and interpretation of the different types of experiments.

Experiments like those in Figure 2A [now Figure 3A] were performed with a widefield fluorescence microscope (one-photon) that does not provide lifetime measurements. As explained in our response to #2 above, we now provide additional RCaMP lifetime measurements in Figure 3—figure supplement 2; these are quite consistent with the conclusions based on the widefield experiments.

4) In Figure 2B – 50% decay times are reported. Is that 50% of the initial spike of Ca? Or is it from a curve fitting to what appears to be a normal exponential decay curve that occurs following the spike. In other words is the decay of mitochondrial Ca biphasic and if so shouldn't both two time constants and their relative contributions be reported?

As explained in our response to #2 above, we now avoid any confusion about kinetics of the mitochondrial Ca^2+^ signals by providing an integrated measurement of the area under the curve. A detailed study of the kinetics of mitochondrial Ca^2+^ handling is beyond the scope of our experiments.

5) In the MCU knockdown mitochondrial NADP(H) signals in Figure 2A the oxidative transients seem smaller than in the controls. That does not seem representative based on Figure 2—figure supplement 3. To avoid confusion would it be possible to use a more representative experiment for Figure 2A?

Thank you for pointing this out. We have now included a more representative experiment.

6) Subsection “Calcium entry into mitochondria is required for strong activation of TCA metabolism”. I think there should also be some discussion of AGC1 activity being stimulated directly by cytosolic Ca^2+^. This is mentioned briefly later in the Discussion but it seems like it could be a significant factor in these studies and in their interpretation.

We have now included a paragraph in the Results, on the possible compensation by MAS (or the glycerol-P shuttle), especially during the activity-associated rise in cytosolic Ca^2+^. Nevertheless, MCU KD has a much larger effect on matrix Ca^2+^ than on cytosolic Ca^2+^, so the matrix-side effect is likely the more important one to mention here (as we do).

In the Discussion, we consider the possible contributions of Ca^2+^ stimulation of AGC1 and of upregulation of MAS in the MCU KD.

7) Subsection “Energy demand from Na^+^ or Ca^2+^ pumping triggers aerobic glycolysis”. Adding strophanthidin in the zero Ca^2+^ shows that ATP consumption stimulates GAPDH activity. However, wouldn't it be also worthwhile to determine if strophanthidin also blocks the cytosolic NADH increase triggered by stimulation when Ca^2+^ is present? In other words can the authors address whether Ca^2+^ does or does not contribute to accelerated glycolysis (independently of increased energy demand)?

This is a complicated question, because blockade of the Na^+^/K^+^ pump is not necessarily sufficient to prevent ATP hydrolysis associated with Ca^2+^ extrusion (through other pathways), and because Na^+^/K^+^ pump inhibition leads rapidly to cell death by swelling under normal ionic conditions. Strophanthidin also causes severe dysregulation of Ca^2+^ and Na^+^, even at low doses. Our future experiments are targeted to address this general conceptual question.

8) Discussion. Please address here whether or not Ca^2+^ regulation of AGC1 (i.e. enhanced MAS activity) also could contribute to elevated NADH in the matrix. (this is a reiteration of my comment #6).

As mentioned in #6, we have discussed this possibility in the Discussion.

9) Discussion: Is the "rapid dip" blocked by strophanthidin?

As mentioned in #6, we have discussed this possibility in the Discussion.

10) The data presented show that Ca^2+^ is little more than a bystander in the regulation of glycolysis. The "distinct roles" formulation of the paper needs to be walked back.

We would not say it is a “bystander”, as eliminating Ca^2+^ influx makes the NADHcyt transients nearly disappear. Ca^2+^ influx plays a major role, but not specifically as a signaling molecule. We have added a better explanation of its apparent role in energy demand, as a surprisingly large contributor (Results).

11) A scheme showing the biochemical linkages and cell-biological relationships (subcellular compartments, etc) under consideration is necessary. Readers should not be made to forage through biochemistry texts and online resources to make sense of the basic questions a paper is attempting to address.

We are happy to provide a scheme of the relevant biochemical processes and their compartmentation, now shown in Figure 1; (the figure supplements show additional features related to the NADH shuttles and ion ATPases; these are included as separate figures to avoid overwhelming complexity).

12) It is important to know the extent of MCU protein depletion; but it is acknowledged that this may be technically difficult. Have all the options been exhausted? Because the extent of MCU depletion is not known, references to possible non-uniporter routes of mitochondrial Ca^2+^ entry (subsection “Calcium entry into mitochondria is required for strong activation of TCA metabolism”) are not only speculative but are potentially provocative.

Thank you for pointing this out. We know the MCU knockdown is quite variable and may be incomplete; we have revised this sentence to mention that remnant MCU may be responsible in the Results. Additional explanations for the initial peak in mitoRCaMP are also included in the Results.

13) The NADHcyto transient: There was a decrease in the NADHcyt transient with MCU depletion that could not be explained by (greater) flux through LDH because it was not "rescued" by the LDH inhibitor. It was stated in the text that the decrease could reflect increased MAS activity; this would constitute a compensation for loss of MCU.

As mentioned in #6, we have added a local explanation (Results) about the possibility of increased/compensatory expression of the mitochondrial shuttles.

14) NAD(P)H dip: There was a small decrease in the amplitude of the NAD(P)H autofluorescence dip in MCU kd slices, and the kinetics of the dip were altered; these changes, while small, were significant (Figure 2—figure supplement 3A).

In the Results, we mentioned that the dip of MCU-KD mice decreases when slices are stimulated with 25 pulses. In the figure legend of now Figure 3—figure supplement 3A, we now mention that the change in the kinetics of the dip likely reflects less overlap between the oxidative and reductive phases of the NAD(P)H signal due to the attenuation of the overshoot.

15) Likewise, there were small but significant changes in the concomitant O_2_ signal, in both the amplitude and kinetics (specifically, in the "time to O_2_ dip" (Figure 2—figure supplement 3B).

We have emphasized the changes in the O_2_ signal in the Results section. In the Discussion section, we now say that the O_2_ dip is slightly diminished or delayed by MCU knockdown (depending on the duration of the stimulus).

FAD+ undershoot during the NAD(P)H overshoot: In MCU kd slices, there was less of a decrease in the FAD+ undershoot compared to decrease in the NAD(P)H overshoot. What does the FAD+ undershoot reflect: PDH activity (the E3 subunit: FADH2 -> FAD+), SDH activity (FAD+ -> FADH2) or the glycerol-3-P-dehydrogenase (FADH2 -> FAD+)? Though the latter describes the NAD(P)H overshoot phase, the fact that there is a different effect on the NAD(P)H overshoot and FAD+ undershoot suggests that the FAD+ signal does not simply reflect TCA FAD+ -> FADH2 (at SDH), and that what contributes to the FAD+ signal may differ between Ctrl and MCU kd. The latter point would support\ that metabolic fluxes in response to stimuli can be different in MCU kd slices.

This is an interesting point. We have now commented on the possibility of the glycerol-P shuttle contributing to the transfer of reducing equivalents into the mitochondria (please see #6). However, we also mention the possibility of glutamate and glutamine sustaining flux through part of the TCA cycle involving SDH, by entering at the α-ketoglutarate level and exiting as oxaloacetate (Discussion).

16) FAD+ undershoot in MCU kd slices likely does not reflect flux through PDH or through SDH (since otherwise less of an attenuation of the NAD(P)H signal would be expected). So what does the FAD+ undershoot reflect in MCU kd? Increased glycerol-3-P-dehydrogenase activity? If yes, then there can be substantial ROS associated with that activity (see Quinlan 2013 PMID: 24024165), which could provoke additional changes.

As commented in #15, there is still the possibility of SDH involvement if glutamate becomes a substrate for the TCA cycle. We have mentioned that the glycerol-P shuttle may also compensate for the reduced TCA cycle activity due to MCU loss (see #6). The relative contributions of these potential pathways, and their consequences for ROS production, are relevant topics for further investigation but are beyond the scope of the present study.

17) Discussion: focus is on emphasizing the effect of MCU kd on the NAD(P)H overshoot and the changes to the dip (NAD(P)H and O_2_) and the FAD+ undershoot seems to be deemphasized. At least in the text of Results, a fuller description of the changes in the dip and undershoot would be needed.

We now mention the unexpected findings on the less attenuated FAD^+^ undershoot in the Results.

18) This dataset might be the first actual evidence that metabolism compensates/responds/is not neutral to MCU loss (beyond the surprising observations in other studies that MCU loss appears to not impact metabolism). Acknowledging this evidence that metabolism is not neutral to MCU loss seems important in light of observations from several studies that MCU loss appears to not alter metabolism, which can lead to the conclusion that MCU plays no role in metabolism.

Thank you for this observation. We hope that the current version of the manuscript emphasizes more the impact of MCU loss on neuronal metabolism.

19) MAS depends on AGC1/2 (aralar/citrin or SLC25A12/SLC25A13) which contain EF-hands that face the IMS and are therefore theoretically exposed to cytosolic Ca^2+^; it would be worth pointing this out, and that the higher cyto Ca^2+^ during the pulses may lead to greater activation of MAS. The same would be the case for glycerol-3-P-dehydrogenase if it were to play a role.

We have addressed this possibility in the previous answers (please see #6, we have discussed this possibility in the Discussion.

20) Referring to the NAD(P)H overshoot as TCA-refilling may be incorrect, and prematurely elevates an observation into an interpretation. That there is an NAD(P)H overshoot suggests that stimulation is followed by a phase in which the mitochondrial matrix is in a relatively reduced state. This may reflect a disproportionate NADH generation by PDH and TCA cycle relative to NADH oxidation by the electron transport chain.

We cannot find the statement on the NAD(P)H overshoot acting as a TCA-refilling event, but perhaps this comment refers to the statement about “a major role for [Ca^2+^]_MITO_ in activating dehydrogenases in the TCA cycle to resupply mitochondrial NADH after neuronal stimulation.” We think this statement agrees with the last sentence of the comment, as indeed we think that the overshoot arises from the imbalance between the “resupply” of NADH encouraged by Ca and the oxidation by the ETC spurred by ATP demand.

The overshoot could reflect a shift in oxidation state of the NADPH pool.

We acknowledge this possibility. However, the NAD(P)H overshoot does behave as expected from TCA-based production of NADH from pyruvate, based on the MPC inhibition. Nevertheless, it could be flux diverted to malic enzyme, or could occur via transhydrogenase (which connects the redox states of NADH and NADPH) – but it does depend on pyruvate and Ca^2+^.

21) About the mixed results in the literature on the effects of mitochondrial Ca^2+^ the different phases of NAD(P)H response to stimulation: That there are "contradictory results" is "merely" mentioned in Results and in Discussion. Has the current study been designed in any way that attempts to better understand the role of mito Ca^2+^ in the NAD(P)H response?

The previous work was done with widely divergent preparations (cultured DRG vs. brain slice) and methods of stimulation (electrical vs. application of kainate or glutamate). Our study is designed to use a wide range of manipulations, ion removal, pharmacologic, and genetic, to alter mitochondrial Ca^2+^. We offer our results to augment rather than supersede the literature on this subject, while the effects on glycolysis are completely novel.

22) Discussion: "We found that oxygen utilization,.…, was not altered by MCU knockdown". This does not seem to be accurate based on the data of Figure 2—figure supplement 3.The sentence "More importantly, Ca^2+^ itself may not be necessary if.… Ca^2+^ extrusion". Metabolic response to Ca^2+^ extrusion and the use of/requirement for Ca^2+^ as part of that metabolic response can be independent.

This sentence has been rewritten and now states that: “More importantly, activity-induced increases in [Ca^2+^]_CYT_ may not be necessary for the metabolic transient, although this increase apparently accounts for a surprisingly large part of the metabolic transient. It may be that the fast glycolytic response is simply reactive to the energy demand resulting from Ca^2+^ extrusion, and that this is a large fraction of the total energy demand produced by activity.”